

# Bayesian Dark Target Algorithm for MODIS AOD retrieval over land

Antti Lipponen[1], Tero Mielonen[1], Mikko R.A. Pitkänen[1,3], Robert C. Levy[2], Virginia R. Sawyer[2], Sami Romakkaniemi[1], Ville Kolehmainen[3], and Antti Arola[1]

[1]Finnish Meteorological Institute, Atmospheric Research Centre of Eastern Finland, Kuopio, Finland
[2]Climate and Radiation Laboratory, NASA Goddard Space Flight Center, Greenbelt, MD, USA
[3]University of Eastern Finland, Department of Applied Physics, Kuopio, Finland

*Correspondence to:* Antti Lipponen (antti.lipponen@fmi.fi)

**Abstract.** We have developed a Bayesian Dark Target (BDT) algorithm for the Moderate Resolution Imaging Spectroradiometer (MODIS) aerosol optical depth (AOD) retrieval over land. In the BDT algorithm, we simultaneously retrieve all pixels in a granule, utilize spatial correlation models for the unknown aerosol parameters, use a statistical prior model for the surface reflectance, and take into account the uncertainties due to fixed aerosol models. The retrieved parameters are total AOD at

550 nm, fine-mode fraction (FMF), and surface reflectances at four different wavelengths (466, 550, 644, and 2100 nm). The accuracy of the new algorithm is evaluated by comparing the AOD retrievals to Aerosol Robotic Network (AERONET) AOD. The results show that the BDT significantly improves the accuracy of AOD retrievals over the operational Dark Target (DT) algorithm. A reduction of about 29 % in the AOD root mean square error and decrease of about 80 % in the median bias of AOD were found globally when the BDT was used instead of the DT algorithm. Furthermore, the fraction of AOD retrievals

inside the $\pm(0.05 + 15\%)$ expected error envelope increased from 55 % to 76 %. In addition to retrieving the values of AOD, FMF and surface reflectance, the BDT also gives pixel-level posterior uncertainty estimates for the retrieved parameters. The BDT algorithm always results in physical, non-negative AOD values, and the average computation time for a single granule was less than a minute on a modern personal computer.

## 1  Introduction

Atmospheric aerosols are small solid or liquid particles suspended in the atmosphere. They have a significant effect on the climate (IPCC, 2013; Kaufman et al., 2002) and they are found to impact, for example, the cloud formation processes, and scattering and absorbtion of solar radiation in the atmosphere. Furthermore, the smallest atmospheric aerosol particles may enter the human lungs when inhaled and may be hazardous to human health (Dockery et al., 1993; Seaton et al., 1995; Pope III et al., 2002; Cohen et al., 2017). As aerosols have widespread climate and health effects, as they may be transported in the

atmosphere very far from their sources, and as the effect of aerosols is one the biggest sources of uncertainty in future climate predictions it is crucial to get accurate information on aerosols. Remote sensing of aerosols using satellite based instruments provides means to globally retrieve aerosol properties.



The Moderate Resolution Imaging Spectroradiometers (MODIS) on board NASA's Terra and Aqua satellites are among the oldest still operating instruments orbiting the Earth and collecting information on Earth's surface and atmosphere. Terra and Aqua are both polar orbiting satellites with wide swaths and they scan the entire surface of the Earth every 1-2 days. An algorithm to retrieve the aerosol properties, such as the aerosol optical depth (AOD), is the Dark Target (DT) which uses

MODIS data measured over dark surfaces (Kaufman et al., 1997a; Levy et al., 2013). There are two different versions of the DT algorithm: one for retrievals over land and another for retrievals over ocean. In this work, we concentrate on the retrievals over land. The physical concept behind the DT algorithm is based on the brightening effect in which the increased amount of aerosols over dark surfaces will reflect more solar radiation back to space and thus will make the scene look brighter. In practice, the retrieval is carried out by finding the aerosol properties that minimize the difference between the top-of-atmosphere (TOA)

reflectances corresponding to radiative transfer simulations and the TOA reflectances measured by the MODIS instrument. One of the biggest problems in this type of approach is to distinguish between the fraction of TOA reflectance that was caused by the aerosols and the fraction that was caused by the land surface (Hyer et al., 2011; Mielonen et al., 2011; Gupta et al., 2016a). In the DT algorithm, surface reflectance at 2.1 $\mu$m is estimated and linear surface reflectance relationships are used to get an estimate for the surface reflectances at shorter wavelengths (466 and 644 nm). The current operational version of the

DT algorithm is the Collection 6 (C6) (Levy et al., 2013). The standard C6 aerosol retrieval products (named MOD04_L2 and MYD04_L2 for Terra and Aqua satellites, respectively) include the AOD and the fraction of fine mode aerosol particles (fine mode fraction, FMF) with pixel resolution of $10 \times 10$ km$^2$ at nadir. The MODIS DT aerosol products are freely and openly available, and are delivered in packages that consist of 5 minutes of measurement data and represent an area of about $2330 \times 2030$ km$^2$. These 5 minute data packages are referred to as granules. The MODIS data can be downloaded for example from

the NASA LAADS DAAC system at https://ladsweb.modaps.eosdis.nasa.gov/.

Another widely used retrieval algorithm for MODIS is the Deep Blue (DB) (Hsu et al., 2004, 2013). The latest version of the algorithm is the C6 DB (enhanced) algorithm. The basic principle of the DB retrieval is similar to DT: find aerosol parameters that minimize the data misfit between the measured and modelled reflectances. In DB, the maximum likelihood principle is used in finding the unknown aerosol parameters. DB is an algorithm for retrievals over land and it was developed

for aerosol retrievals especially over bright-reflecting surfaces. The capability of retrieving aerosol properties over bright-reflecting surfaces is useful for example in retrieving dust properties over deserts. DB does not carry out retrievals over snow or ice. The DB uses various MODIS spectral bands for cloud screening and aerosol typing, and the bands centered at 412, 490, and 670 nm are used for the actual retrieval. For some surface types DB uses similar surface reflectance relationships as DT, and for some surface types the surface reflectance values are directly taken from a database. The DB MODIS retrievals are

delivered with the same C6 MODIS aerosol products as the DT retrievals.

Both the DT and DB carry out the retrieval pixel by pixel. This means every pixel is retrieved independently of each other. This pixel by pixel approach makes the algorithm computationally efficient. Often, however, aerosol properties have strong spatial correlations. Taking advantage of the spatial correlations of aerosol properties in the retrieval can, in many cases, improve the accuracy of the retrieved parameters. One of the largest error sources in the MODIS AOD retrieval is the (partially)

unknown surface reflectance: typical error for the retrieved AOD is proportional to ten times the error in estimated surface





reflectance (Kaufman et al., 1997b). If more accurate surface reflectance values were used it could improve the accuracy of the retrieval. Furthermore, one increasingly important problem with DT is that it sometimes retrieves unphysical negative AOD values. As the MODIS instruments have already passed their designed lifetimes and their sensitivities are rapidly decreasing, they require more and more frequent calibrations. As a result of sensor degradation and frequent calibrations, the number of

negative AOD retrievals with the DT algorithm is increasing.

    In this work, we developed a Bayesian Dark Target (BDT) aerosol retrieval algorithm for MODIS aerosol retrieval over land. The new algorithm is based on the DT algorithm and the inversion part of the algorithm is reformulated as a statistical (Bayesian) inverse problem (Kaipio and Somersalo, 2005; Calvetti and Somersalo, 2007; Gelman et al., 2014). While the DT retrieves one pixel at a time, in the BDT all the dark surface and cloud-free pixels of a granule are retrieved simultaneously.

BDT allows the use of statistical prior models for the unknown parameters. The prior models are probability distribution models for prior information such as ranges of feasible values of the parameters and spatial correlations. BDT also allows us to take into account the statistics of the measurement noise and compensate for model uncertainties caused, for example, by the fixed aerosol models. Instead of the surface reflectance relationships used in the DT algorithm, we include the surface reflectances at different wavelengths as unknown parameters and retrieve the actual surface reflectances simultaneously with the aerosol

properties.

## 2   Bayesian Dark Target Algorithm

MODIS aerosol products retrieved using the DT are among the most widely used aerosol products. The MODIS C6 standard aerosol products include the retrieved aerosol properties and measurement data with spatial resolution of about $10 \times 10$ km$^2$ at nadir. In DT, the retrieval is carried out separately for each pixel and the retrieval parameters are the total AOD at 550 nm $\tilde{\tau}$, fine

aerosol model weighting $\eta$ (fine mode fraction, FMF), and the surface reflectance at $2.1\mu$m $\rho_{2.1\mu\mathrm{m}}^s$. The surface reflectances at shorter wavelengths are estimated using predefined linear surface reflectance relationships that depend on the normalized difference vegetation index (NDVI) at short-wave infrared (SWIR) and the scattering angle of the light (Remer et al., 2001; Levy et al., 2007). In the DT retrieval, the TOA reflectances are simulated by mixing the reflectances corresponding to two different aerosol models:

$$\tilde{\rho}^{\mathrm{TOA}} = \eta\tilde{\rho}^{\mathrm{TOA,fine}} + (1-\eta)\tilde{\rho}^{\mathrm{TOA,coarse}} \tag{1}$$

where $\tilde{\rho}^{\mathrm{TOA}}$ denotes the simulated TOA reflectances, $\eta$ is the FMF, and $\tilde{\rho}^{\mathrm{TOA,fine}}$ and $\tilde{\rho}^{\mathrm{TOA,coarse}}$ denote the TOA reflectances simulated according to the fine and coarse aerosol models, respectively. There are three different fine aerosol models, one coarse (dust) aerosol model, and one continental aerosol model in DT. The TOA reflectances and other radiative transfer related variables corresponding to each aerosol model are precomputed and stored in lookup tables (LUT) to make the algorithm

computationally more efficient. In the DT retrieval, the fine aerosol model to be used is taken from a predefined database that contains aerosol model information based on location and season. For more information on the C6 DT retrieval algorithm see, for example, Levy et al. (2013).





BDT is a retrieval algorithm based on DT. In BDT, the inversion part of the DT algorithm is formulated in a statistical (Bayesian) framework. In this statistical framework, the solution to the inverse retrieval problem is not a single value but a posterior probability distribution model of the unknown parameters given the measured MODIS TOA reflectances and prior information that we have on the unknowns. As the complete statistical model of the problem is the posterior probability distri-
bution, it allows us to derive single point estimates that are referred to as the retrievals, and quantify the posterior uncertainties of the retrievals for each pixel. The statistical framework also allows us, for example, to utilize information about the measurement noise and use data from as many MODIS spectral bands as available for the retrieval. In the BDT algorithm:

- We use data from MODIS bands 3 (466 nm), 4 (552 nm), 1 (644 nm), and 7 (2.1 $\mu$m). All other bands could be used as well but four bands are selected to keep the computational costs moderate.

- We retrieve the total AOD at 550nm, the FMF, and the surface reflectances at 4 MODIS bands.

- The surface reflectances at all bands are simultaneously retrieved with AOD and FMF. The surface reflectance relationships that are used in DT are not needed.

- We simultaneously retrieve all unknown parameters in all pixels of a granule.

- We use prior probability density models for the values and the spatial correlation structure of the unknowns. The prior
probability density models are used to encode the prior knowledge such as spatial correlation information, seasonal variability or positivity constraints into the retrieval.

- We utilize an approximation error model for the model uncertainties in the simulated TOA reflectances caused by the uncertainties in the aerosol models and radiative transfer simulations.

In the BDT AOD retrieval, statistical prior models for the retrieved parameters can be used. We make the following modelling
selections in the BDT:

- To avoid negative AOD retrievals, we retrieve AOD in logarithmic scale $\tau = \log(\tilde{\tau} + 1)$.

- Instead of TOA reflectances $\tilde{\rho}^{\mathrm{TOA}}$ in linear scale, we write also the TOA reflectances in the models in logarithmic scale as $\rho^{\mathrm{TOA}} = \log(\tilde{\rho}^{\mathrm{TOA}} + 1)$.

- We model all unknown parameters in a granule by multivariate Gaussian prior models. The prior models are fully
described by their expected value vectors and covariance matrices:

- AOD $\boldsymbol{\tau} \sim \mathcal{N}(\mathbb{E}_{\boldsymbol{\tau}}, \boldsymbol{\Gamma}_{\boldsymbol{\tau}})$

- FMF $\boldsymbol{\eta} \sim \mathcal{N}(\mathbb{E}_{\boldsymbol{\eta}}, \boldsymbol{\Gamma}_{\eta})$

- Surface reflectances $\boldsymbol{\rho}^s \sim \mathcal{N}(\mathbb{E}_{\boldsymbol{\rho}^s}, \boldsymbol{\Gamma}_{\boldsymbol{\rho}^s})$

- We model AOD, FMF, and surface reflectances at all bands as mutually uncorrelated variables.





    – We model the observation noise, and the approximation errors in TOA reflectances due to aerosol and radiative transfer models as additive multivariate Gaussian random variable $e$ with distribution $e \sim \mathcal{N}(\mathbb{E}_e, \mathbf{\Gamma}_e)$

In the BDT, we look for the *maximum a posteriori* (MAP) estimate for the unknown parameters. The prior and likelihood models that are used in the construction of the posterior model are explained in more detail in Section 3. With the models selected, the MAP estimate can be computed as

$$(\boldsymbol{\tau}, \boldsymbol{\eta}, \boldsymbol{\rho}^s)_{\mathrm{MAP}} = \arg \min_{\boldsymbol{\tau}, \boldsymbol{\eta}, \boldsymbol{\rho}^s} \left( \left\| \mathbf{L}_e \left( \boldsymbol{\rho}^{\mathrm{TOA,MODIS}} - f(\boldsymbol{\tau}, \boldsymbol{\eta}, \boldsymbol{\rho}^s; \boldsymbol{\gamma}) - \mathbb{E}_e \right) \right\|^2 + \left\| \mathbf{L}_{\boldsymbol{\tau}} \left( \boldsymbol{\tau} - \mathbb{E}_{\boldsymbol{\tau}} \right) \right\|^2 + \left\| \mathbf{L}_{\boldsymbol{\eta}} \left( \boldsymbol{\eta} - \mathbb{E}_{\boldsymbol{\eta}} \right) \right\|^2 + \left\| \mathbf{L}_{\boldsymbol{\rho}^s} \left( \boldsymbol{\rho}^s - \mathbb{E}_{\boldsymbol{\rho}^s} \right) \right\|^2 \right)$$

(2)

where $\boldsymbol{\tau} = \log(\tilde{\tau} + 1)$ is the (logarithm) of AOD at 550nm, $\boldsymbol{\eta}$ denotes the FMF, and $\boldsymbol{\rho}^s$ are the surface reflectances at all bands, respectively, and $\boldsymbol{\gamma}$ denotes auxiliary (fixed) model parameters such as measurement geometry, surface elevation, and aerosol models. $\mathbf{L}_e$, $\mathbf{L}_{\boldsymbol{\tau}}$, $\mathbf{L}_{\boldsymbol{\eta}}$ and $\mathbf{L}_{\boldsymbol{\rho}^s}$ denote the Cholesky factors of $\mathbf{\Gamma}_e^{-1}$, $\mathbf{\Gamma}_{\boldsymbol{\tau}}^{-1}$, $\mathbf{\Gamma}_{\boldsymbol{\eta}}^{-1}$ and $\mathbf{\Gamma}_{\boldsymbol{\rho}^s}^{-1}$, respectively. $f(\boldsymbol{\tau}, \boldsymbol{\eta}, \boldsymbol{\rho}^s; \boldsymbol{\gamma}) = \log \left( \tilde{f}(\boldsymbol{\tau}, \boldsymbol{\eta}, \boldsymbol{\rho}^s; \boldsymbol{\gamma}) + 1 \right)$, $\tilde{f}$ is the observation model based on aerosol and radiative transfer models and is based on LUTs. $\boldsymbol{\rho}^{\mathrm{TOA,MODIS}} = \log(\tilde{\boldsymbol{\rho}}^{\mathrm{TOA,MODIS}} + 1)$ and $\tilde{\boldsymbol{\rho}}^{\mathrm{TOA,MODIS}}$ contains the actual TOA reflectances measured by the MODIS instrument. In our implementation of BDT, we use the L-BFGS-B optimization algorithm (Byrd et al., 1995) to solve the retrieval optimization problem. For further details of the optimization problem, see Appendix A.

To quantify the uncertainties corresponding to the retrieved parameters we can compute an approximation for the posterior covariance matrix as:

$$\mathbf{\Gamma}_{\boldsymbol{\tau}, \boldsymbol{\eta}, \boldsymbol{\rho}^s} \approx \left( \mathbf{\Gamma}_{pr}^{-1} + \mathbf{J}^{\mathrm{T}} \mathbf{\Gamma}_e^{-1} \mathbf{J} \right)^{-1}$$

(3)

where the block diagonal matrix $\mathbf{\Gamma}_{pr} = \mathrm{diag}\left(\mathbf{\Gamma}_{\boldsymbol{\tau}}, \mathbf{\Gamma}_{\boldsymbol{\eta}}, \mathbf{\Gamma}_{\boldsymbol{\rho}^s}\right)$, and $\mathbf{J} = [\partial f / \partial \tau, \partial f / \partial \eta, \partial f / \partial \rho^s]$ is the Jacobian matrix evaluated at the MAP estimate. The diagonal of the posterior covariance matrix contains posterior variances of each retrieved parameter at each pixel.

## 3    Bayesian Dark Target Models

### 3.1    Prior models

Prior probability density models are used in the BDT retrieval to model information we have on unknown parameters prior to the retrieval. In the BDT, we use Gaussian prior models augmented with constraints that exclude non-physical solutions. For example, for the FMF the retrieval is restricted to an interval between 0-1. The multivariate Gaussian prior models are defined by their expected value vector and covariance matrix. In aerosol retrievals, the expected value vectors for aerosol parameters can be constructed, for example, by using values from aerosol climatologies. Covariance matrices encode information on the prior uncertainty of the parameters and correlations between different pixels.



**Table 1.** The covariance function parameters used in aerosol optical depth (AOD) and fine mode fraction (FMF) prior models.

| Covariance function parameter | AOD value | FMF value |
|---|---|---|
| Correlation range $r_{\mathrm{range}}$ | 50 km | 50 km |
| Nugget $\sigma^2_{\mathrm{nugget}}$ | $2.5 \cdot 10^{-3}$ | 0.01 |
| Sill $\sigma^2_{\mathrm{sill}}$ | 0.10 | 0.25 |
| $p$ | 1.5 | 1.5 |

### 3.1.1 Prior model for the AOD

In the BDT algorithm, the AOD is retrieved on a logarithmic scale to avoid negative AOD retrievals and multivariate Gaussian distributions are used as the prior models for the logarithm of the AOD. The expected value vector for AOD is based on the MAC-V2 climatology by Kinne et al. (2013). The MAC-V2 climatology contains monthly AOD values in 1 degree by 1 degree grid. In the BDT retrieval, the nearest value from the MAC-V2 climatology is taken as the prior expectation for each pixel to be retrieved.

The spatial correlations and variances in the logarithm of AOD are modelled by using a covariance function that defines the AOD covariance matrix as:

$$\mathbf{\Gamma}_{\boldsymbol{\tau}}(i,j) = \sigma^2_{\mathrm{nugget},\tau}\delta_{i,j} + \sigma^2_{\mathrm{sill},\tau} \exp\left\{-3\left\|\frac{\boldsymbol{x}_i - \boldsymbol{x}_j}{r_{\mathrm{range},\tau}}\right\|^p\right\} \tag{4}$$

where $\Gamma_\tau(i,j)$ is the $(i,j)$ element of the prior covariance matrix $\mathbf{\Gamma}_{\boldsymbol{\tau}}$, $\delta_{i,j}=1$ when $i=j$ and $\delta_{i,j}=0$ when $i\neq j$, and $\|x_i - x_j\|$ denotes the distance between the pixels $i$ and $j$. $\sigma_{\mathrm{nugget},\tau}$ denotes the so-called nugget and it represents the local component of the AOD variance (no spatial correlation). The sill $\sigma_{\mathrm{sill},\tau}$ describes the variance related to the spatially correlated component of AOD. Consequently, the total variance of AOD $\sigma^2_\tau = \sigma^2_{\mathrm{nugget},\tau} + \sigma^2_{\mathrm{sill},\tau}$. The correlation range $r_{\mathrm{range},\tau}$ and $p_\tau$ define the spatial correlation length and smoothness of the AOD fields. The larger the selected correlation range is, the larger the spatial structures we expect to see in AOD. In BDT, we used fixed values for the covariance function parameters and they are listed in Table 1. The sill and nugget parameter values were selected by analyzing previous MODIS retrievals. The range value was selected as 50 km. This selection was made to let the neighbouring pixels have relatively high spatial correlation but on the other hand to allow for certain features such as smoke plumes to be retrieved correctly and not be smoothed out. The term $p_\tau$ was selected as 1.5 based on visual inspection of retrieved AOD fields. In this version of BDT, the covariance function parameters were manually selected but it is also possible to infer the covariance function parameters, for example, by performing variogram analysis on previous AOD retrieval data similarly as, for example, in Chatterjee et al. (2010). This type of spatial correlation modelling is often used in geostatistical methods such as kriging.





### 3.1.2 Prior model for the FMF

For the FMF, we use similar Gaussian prior as for the AOD. The prior expectation value for FMF is taken from the MAC-V2 climatology as for the AOD. The FMF is modelled as a spatially correlated parameter and the same type of covariance function as for the AOD is used to construct the prior covariance matrix $\Gamma_\eta$. The range, sill and nugget values for the FMF prior model covariance are listed in Table 1. The sill was intentionally selected as relatively large value to allow for high prior uncertainty in the spatial part of the prior model.

### 3.1.3 Prior model for the surface reflectance

In the BDT algorithm, the surface reflectances at different wavelengths are treated as unknown parameters and they are simultaneously retrieved with AOD and FMF. In the BDT algorithm, we use Gaussian prior models for the surface reflectances. We model the surface reflectances at different bands as uncorrelated and the surface reflectances at each band as spatially uncorrelated. We note that this selection may not result in the best possible retrieval accuracy but makes the processing of large number of MODIS granules significantly faster than with correlated models. With these choices for the surface reflectance, the prior model becomes an uncorrelated Gaussian density which is described by the expected surface reflectance values and their variances at each pixel. As expected values for the surface reflectance, we use the MODIS MCD43C3 albedo product blue sky albedos computed with the weighting coefficient 0.5 (50% of the white sky albedo and 50% of the black sky albedo). The daily MODIS albedo product is stored in 0.05 degree by 0.05 degree grid. For the BDT, we precompute monthly expected surface reflectance corresponding to the surface albedo product grid. The monthly surface reflectance is computed as the temporal average of surface reflectances ± 45 days around the middle day of the month. In the retrieval, the expected values for the surface reflectances are computed as an average of 3 closest pixels in the monthly surface reflectance. Both the temporal variance in the original surface albedo product and the variance due to averaging are taken into account in the construction of the surface reflectance variance.

In near real-time analysis, the surface reflectance product for the retrieval day is not necessarily available. Therefore in the construction of the surface reflectance prior model, we used the MODIS albedo products corresponding to the retrieval month one year before the retrieval. This way it is possible to evaluate the near real time retrieval performance of the algorithm.

## 3.2 Observation model

In the DT algorithm, the TOA reflectance $\rho^{\mathrm{TOA,MODIS}}$ measured by MODIS is modelled according to Equation (1) as a mixture of reflectances produced by two aerosol models: one for fine and one for coarse aerosols. The TOA reflectance corresponding to Lambertian surface, an aerosol model, and one MODIS band is computed as

$$\rho_\lambda^{\mathrm{TOA}}(\theta_0,\theta,\phi) = \rho_\lambda^a(\theta_0,\theta,\phi) + \frac{T_\lambda(\theta_0)T_\lambda(\theta)\rho_\lambda^s(\theta_0,\theta,\phi)}{1 - s_\lambda\rho_\lambda^s(\theta_0,\theta,\phi)} \tag{5}$$





where $\theta_0$, $\theta$, and $\phi$ are the solar zenith, view zenith and relative azimuth angles, respectively, $\rho_\lambda^a$ denotes the atmospheric path reflectance, $T_\lambda(\theta_0)$ and $T_\lambda(\theta)$ denote the downward and upward atmospheric transmissions, $s_\lambda$ is the atmospheric backscattering ratio, and $\rho_\lambda^s$ the surface reflectance corresponding to a band centered at wavelength $\lambda$.

To make the retrieval algorithm computationally efficient, the values of $\rho_\lambda^a$, $T_\lambda$, and $s_\lambda$ for various measurement geometries and AODs are precomputed into a LUT. Each aerosol model has their own LUT and the fine aerosol model to be used in the retrieval is predefined for each location and season. In the BDT retrieval, we use the same aerosol models as in the DT retrieval. In certain conditions, DT uses continental aerosol as the only aerosol model. If continental aerosol model was selected by the DT (Procedure B in MODIS DT over land retrieval), we use the continental aerosol model as the fine aerosol model and compute the total TOA reflectance as a mixture of TOA reflectances caused by the continental and coarse aerosol models.

Before the DT retrieval is carried out, the LUTs are prepared for the retrieval. The LUT models are first interpolated to the fixed measurement geometry, and then corrected for the surface elevation. In the retrieval, the LUT models are then evaluated by linearly interpolating the values as function of total AOD. In BDT, we use the same LUTs (for 4 different bands) as in the DT. While the DT algorithm uses piecewise linear interpolation, in BDT we use fifth order polynomial interpolation of the LUTs, in order to make the model differentiable wrt. the unknown AOD at all points. The differentiability is required as the retrieval is carried out by solving an optimization problem using gradient based methods.

In the BDT algorithm, the random observation noise in MODIS observations is modelled by an additive noise process

$$
\begin{aligned}
\rho^{\mathrm{TOA}} &= \eta \rho^{\mathrm{TOA,fine}} + (1-\eta)\rho^{\mathrm{TOA,coarse}} + n \\
&= \tilde{f}(\tau, \eta, \rho^s; \gamma) + n
\end{aligned}
\tag{6}
$$

where $n$ denotes the observation noise and $\tilde{f} = \tilde{f}(\tau, \eta, \rho^s; \gamma)$ is the observation model. In BDT, the observation noise is modelled as Gaussian zero-mean random variable, and its variances are based on MODIS aerosol product variable `STD_Reflectance_Land`.

## 3.3 Approximation errors

In statistical (Bayesian) retrieval framework, it is possible to model and take into account the uncertainties and inaccuracies related to the physical models that are used in the retrieval (both aerosol and radiative transfer models). The model uncertainties can be related, for example, to uncertainty in the values of the auxiliary model parameters such as measurement geometry and fixed aerosol models. In the field of statistical inverse problems, these model errors are often referred to as approximation errors (Kaipio and Somersalo, 2007). In the BDT algorithm, we incorporate approximation errors due to fixed aerosol models and inaccuracies in the radiative transfer models. The approximation error is modelled as additive Gaussian random variable $u$. Adding $u$ into the observation model model (6) results in observation model of the form:

$$
\begin{aligned}
\rho^{\mathrm{TOA}} &= \tilde{f}(\tau, \eta, \rho^s; \gamma) + n + u \\
&= \tilde{f}(\tau, \eta, \rho^s; \gamma) + \tilde{e}
\end{aligned}
\tag{7}
$$



where $\tilde{e} = n + u$ includes both the observation noise and model uncertainties. The realization of $u$ is unknown. The objective in the approximation error approach is to marginalize the posterior model with respect to the overall observation error. Typically, an approximate marginalization is obtained by using Gaussian model for $n$ and $u$, leading to the data misfit form in Equation (2) where $\mathbb{E}_e$ and $\mathbf{\Gamma}_e$ are the mean and covariance of the overall error. For details, see (Kolehmainen et al., 2011; Kaipio and Kolehmainen, 2013).

In this study, the estimation of the mean $\mathbb{E}_u$ and covariance $\mathbf{\Gamma}_u$ for the Gaussian approximation error model is carried out by comparing collocated MODIS TOA reflectances with simulated TOA reflectances using AOD and FMF values from AERONET (Holben et al., 1998) observations (for details, see Appendix B). We model the approximation error $u$ spatially but not spectrally as uncorrelated meaning the correlations between the MODIS bands are taken into account. The approximation error statistics are precomputed for different regions and months to account for the spatial and seasonal variations. Similarly as for the surface reflectance model, the approximation error models are constructed using AERONET and MODIS data collected one year before the retrieval month to make the evaluation of the near real time performance of the algorithm possible.

In BDT retrieval, we model the observation noise $n$ and model uncertainties $u$ as mutually uncorrelated and therefore in our model $e = n + u$ is distributed as $e \sim \mathcal{N}(\mathbb{E}_n + \mathbb{E}_u, \mathbf{\Gamma}_n + \mathbf{\Gamma}_u)$.

## 4 Evaluation Of The Algorithm

To test the performance of the BDT algorithm, all MODIS daytime granules of the year 2015 are used. We retrieve all granules from Terra and Aqua (MOD04_D3 and MYD04_D3) and compare the retrievals to AERONET observations (version 3, level 1.5). In the AERONET collocation we follow similar comparison protocol as in Petrenko et al. (2012). That is, we require at least 3 MODIS pixels within 25 km from the AERONET station and at least two AERONET observations within the $\pm 30$ minutes from the satellite overpass. We carry out two comparisons between retrievals with different algorithms:

1. To compare the overall performance and to make the comparison fair between different algorithms, we compare all pixels in which the retrieval was carried out regardless of the DT quality assurance (QA) information of the retrieval.

2. To study how the DT QA information affects the retrievals, we carry out another comparison in which we use the DT and BDT retrievals only at the pixels with DT QA flag 3.

As we evaluate the near real time performance, we use the surface reflectance prior models and the uncertainty models that were constructed using MODIS and AERONET data from 2014 (one year before the test year 2015).

The variables we compare are the AOD at 550 nm and Ångström exponent. AERONET AOD at 550 nm is derived using the Ångström power law and AERONET Ångström exponent (440-675 nm). The AEs are used in the comparison instead of the FMF because

– FMF in the DT algorithm is actually the weighting coefficient between the TOA reflectances corresponding to fine and coarse aerosol models and do not necessarily correspond to true size distribution information





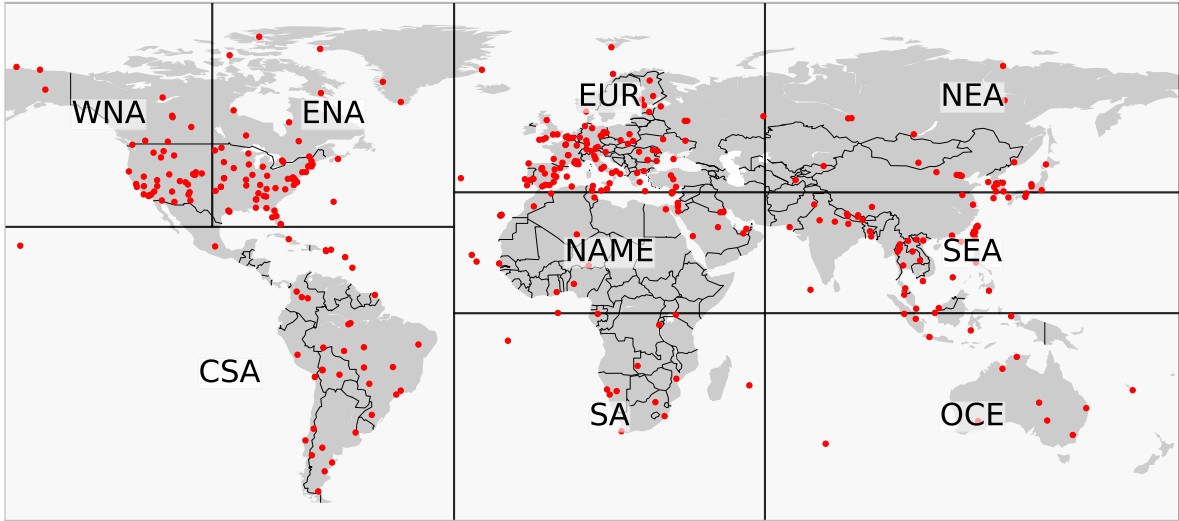

**Figure 1.** Regions used in the evaluation of the algorithm: West North America (WNA), East North America (ENA), Central and South America (CSA), Europe (EUR), North Africa and Middle East (NAME), South Africa (SA), Northeast Asia (NEA), Southeast Asia (SEA), and Oceania (OCE). The red dots show positions of the AERONET stations used in the comparisons.

- In the DT aerosol models, fine aerosol model includes a small amount of coarse particles in it and coarse aerosol model includes a small amount of fine particles in it

- It is ambiguous to derive AERONET based FMF as there are multiple size distribution related products that are based on slightly different algorithms and definitions

- It is possible to derive AE from MODIS retrieval using the aerosol models, retrieved total AOD and FMF, and the AE is also available in the AERONET Direct Sun algorithm outputs

The metrics we use to evaluate the retrieval algorithm performance and compare the MODIS and AERONET retrievals are: correlation coefficient $R$, median bias, and root mean squared (RMS) error. In addition, for AOD we also use the fraction of retrievals inside the DT expected error envelope $\pm(0.05 + 15\%)$ that is we compute the fraction of MODIS AOD retrievals

10  $\tau_{\mathrm{MODIS}}$ that fulfill $0.85\tau_{\mathrm{AERONET}} - 0.05 \leq \tau_{\mathrm{MODIS}} \leq 1.15\tau_{\mathrm{AERONET}} + 0.05$ where $\tau_{\mathrm{AERONET}}$ denotes the AERONET AOD. To get an idea of regional performance of the algorithm, we evaluate the algorithm in 9 different regions. The map of the regions and AERONET stations used for the evaluation is shown in Figure 1. In addition, we also evaluate the retrieval algorithms over urban areas by comparing the retrievals over 17 selected AERONET stations that are located in urban areas. We also carry out a comparison between the BDT and the DB retrievals. In addition, we evaluate the BDT posterior uncertainty estimates by

15  comparing them to the discrepancies between AERONET and BDT algorithm AODs.





## 5   Results

### 5.1   Examples of single granule retrievals

Figure 2 shows AOD and AE retrievals near the Beijing area, China, on 11 October 2015 computed both with DT and BDT. The figure shows clearly that DT overestimates the AOD over the cities of Beijing and Tianjin. The overestimation may be caused

by the urban surface that probably is not well described by the DT surface reflectance relationships used in the operational retrieval (Gupta et al., 2016b). The overestimation of AOD over urban areas due to surface may cause significant biases, for example, to the results of satellite-based air quality studies. In BDT, the AOD retrievals match the AERONET AODs well and cities of Beijing and Tianjin are not visible as high AOD areas in the figure. Furthermore, the DT AE retrievals over Beijing show AE values lower than one indicating large aerosol particles. The AERONET, however, shows AE larger than one

indicating small aerosol particles. BDT shows AE values larger than one for almost all pixels shown in the figure.

Figure 3 shows AOD and AE retrievals over the USA on 10 July 2015. A smoke plume is clearly visible in the figure. In this case, both the DT and BDT produce similar AOD retrievals. Regardless of the spatial correlation model used for AOD in BDT, the plume is not oversmoothed and shows similar details as the DT retrieval. In the BDT AE retrievals, the AE is larger than one in almost all pixels shown in the figure indicating presence of small aerosol particles. In the DT AE retrieval, some

pixels have AE values smaller than one showing presence of large aerosol particles. Large aerosol particles (small AE values) are not, however, typical for this area and season and therefore the small AE values indicating large aerosol particle size seen in the DT data are likely artefacts caused by the retrieval algorithm.

### 5.2   Global performance of the algorithm

The global performance of the algorithm was evaluated using all the daytime retrievals from the year 2015. Figure 4 shows

a global scatter density histogram comparison of the AERONET AOD and retrievals carried out with the DT, BDT, and DB algorithms. Figure 4 was constructed using all retrieved pixels regardless of the quality assurance values. It should be noted that the DT based algorithms (DT and BDT) and DB algorithm apply different pre-processing of the data and the pixels in which the retrieval is carried out is selected differently in these algorithms. The DB algorithm was designed to be able to retrieve AOD also over bright-reflecting surfaces where the DT algorithm may not be used. Therefore, the DB algorithm usually accepts more

pixels for retrieval than the DT algorithm. In this study, the number of AERONET-DB collocations ($N$=57308) was larger than the number of AERONET-DT collocations ($N$=45240). As BDT retrieves the same pixels as the DT algorithm there was no difference in the amount of data between these two retrieval algorithms. It should also be noted that the DT pixels are not necessary a subset of the DB pixels and in some granules the DT and DB pixels may be completely separate sets.

The results show that the BDT AOD retrievals are significantly more accurate than the corresponding DT or DB retrievals

when compared to the AERONET AOD. The fractions of retrievals inside the DT expected error (EE) envelope ($\pm(0.05 + 15\%)$) are 75.7%, 54.6%, and 64.6% for BDT, DT, and DB, respectively. Furthermore, the median absolute errors are about 40% and 20% smaller in BDT than in the DT and DB retrievals, respectively. Also the reduction in the median bias is significant: median biases for BDT, DT, and DB algorithms are 0.009, 0.046, and 0.020, respectively. The feature of both the BDT and





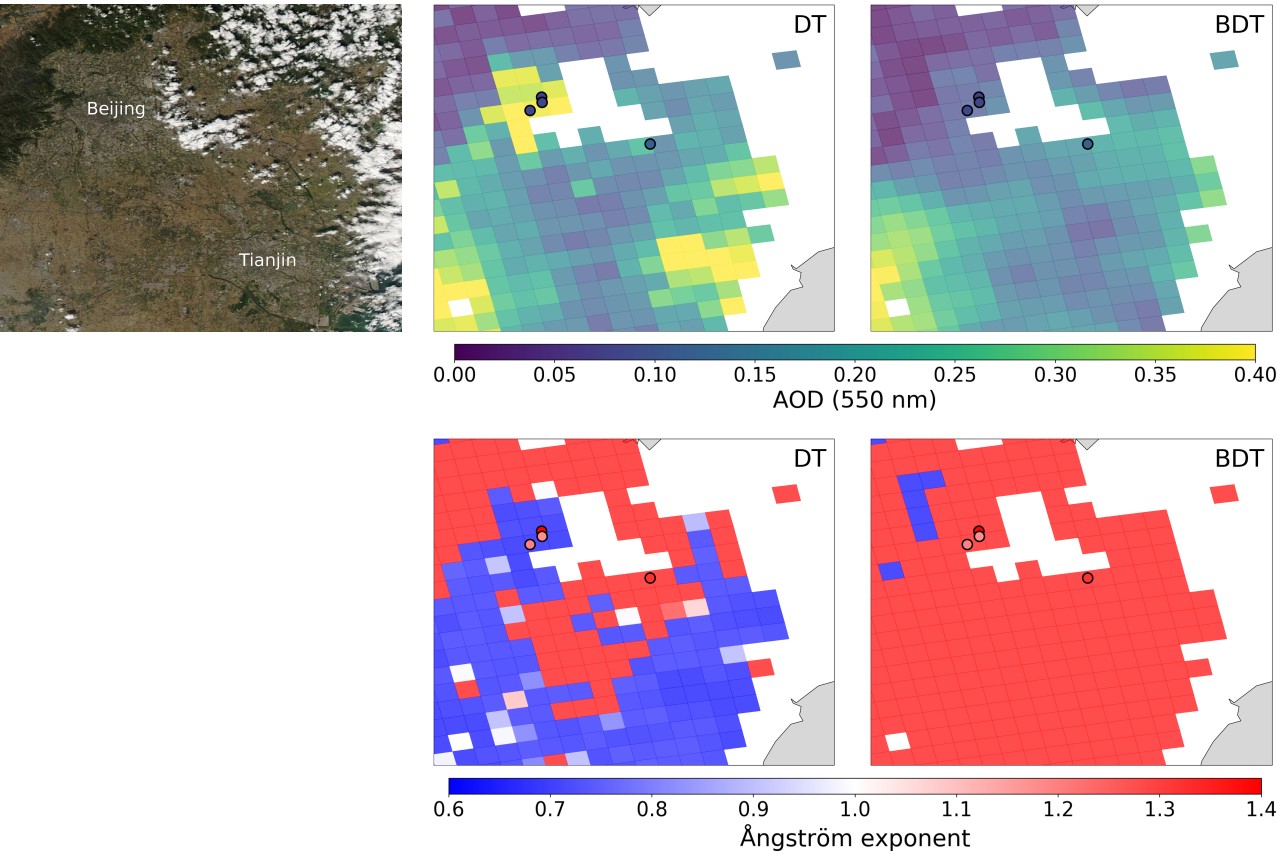

**Figure 2.** Top row: True color image of MODIS Aqua overpass over Beijing area, China on 11 October 2015 (left), AOD retrievals computed with DT (middle) and BDT (right) algorithms. Bottom row: The Ångström exponent retrievals computed with DT (middle) and BDT (right) algorithms. The circles correspond to AERONET AOD and Ångström exponent values at the satellite overpass time.

DB retrievals that they do not allow for negative AOD retrievals is also visible in the figure. There are also clearly more AOD retrievals above the DT EE envelope than below it with all of the algorithms but in the BDT the relative difference between the amount of retrievals above and below the envelope is the smallest.

Figure 5 shows similar plot as Figure 4 but here the comparison was carried out using only the DT and BDT algorithms
5    and pixels with DT QA flag 3 (Levy et al., 2013) for both algorithms. The results were slightly improved for both algorithms when compared with the all pixel retrievals. Even though the difference between the performance of the algorithms is reduced the BDT retrievals are clearly better than the DT retrievals also in this case. This is the result regardless of the filtering of the data that was carried out based on the DT algorithm QA flag and was designed particularly to discard DT pixels with poor quality. The filtering reduced the amount of AERONET collocations by about 40%. The results suggest that the BDT is not
10    only capable of retrieving AOD with significantly improved accuracy than the DT retrieval but also capable of producing good quality retrievals over significantly larger areas.



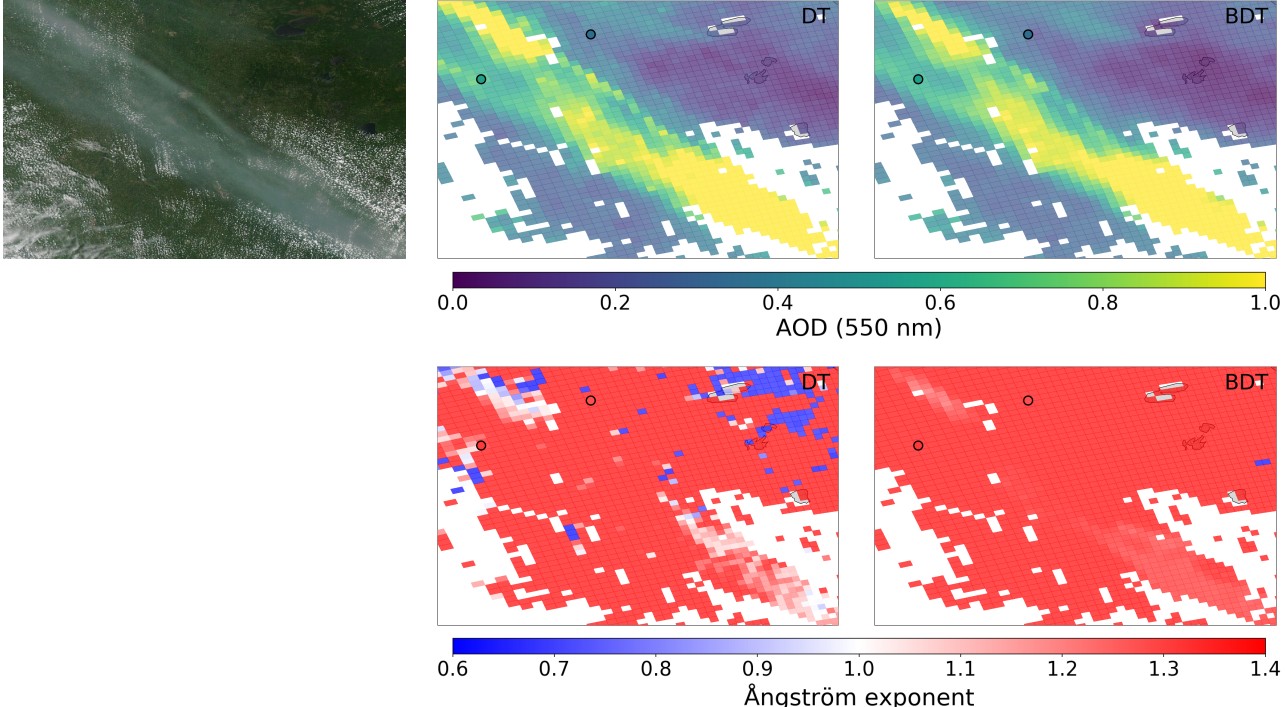

**Figure 3.** Top row: True color image of MODIS Aqua overpass near the border area of Minnesota, and North and South Dakota, USA on 10 July 2015 (left), AOD retrievals computed with DT (middle) and BDT (right) algorithms. Bottom row: The Ångström exponent retrievals computed with DT (middle) and BDT (right) algorithms. The circles correspond to AERONET AOD and Ångström exponent values at the satellite overpass time.

The results for global AE retrievals for the DT and BDT algorithms are shown in Figure 6. If AOD is very small, the reflectances observed by MODIS contain only a very small amount of information about the aerosol size distributions. Therefore, to evaluate the algorithm capability to retrieve size distribution information, we carried out the AE comparison only with retrievals that correspond to AERONET AODs larger than 0.2. The results in this figure include all retrieved pixels. The correlation coefficient is slightly better in DT AE (0.359) than in BDT AE (0.354) retrievals but the difference is negligible. The median and mean absolute errors and the median bias, however, are smaller in BDT retrievals. Visually inspecting the BDT retrievals are better concentrated around the one-on-one line in the scatter plot whereas a large portion of DT retrievals are concentrated around the AE value of about 0.6.

We also evaluated the effect of using the approximation error model and spatial correlation models in the retrieval. The retrievals were carried out in all granules in year 2015 with and without the approximation error model and with and without the spatial correlation models for the AOD and FMF. In the retrievals without spatial correlation models, we set the off-diagonal elements of the prior covariance matrices as zeros both for AOD and FMF. The results are shown in Table 2 and Table 3. The results show that the approximation error model plays the most significant role in improving the retrieval accuracy. Globally,





**Table 2.** Global statistics of AOD retrievals for Bayesian Dark Target (BDT) run with different models. The models considered are the approximation error model and the spatial correlation model for AOD and FMF. X and - in the table indicate that the corresponding model was and was not included in the retrieval, respectively. All pixels were considered in the retrieval and each row correspond to data from 346 AERONET stations and 45240 collocated observations.

| Approximation error model | Spatial correlation model for AOD and FMF | $R$ | Median Bias | $f$ Within $EE_{DT}$ | RMS Error |
|---|---|---|---|---|---|
| X | X | 0.92 | 0.01 | 0.76 | 0.10 |
| X | - | 0.93 | 0.01 | 0.77 | 0.09 |
| - | X | 0.87 | -0.01 | 0.62 | 0.12 |
| - | - | 0.87 | -0.01 | 0.63 | 0.12 |
| DT algorithm, all pixels | | 0.89 | 0.05 | 0.55 | 0.14 |

**Table 3.** Global statistics of Ångström exponent retrievals for Bayesian Dark Target (BDT) run with different models. The models considered are the approximation error model and the spatial correlation model for AOD and FMF. X and - in the table indicate that the corresponding model was and was not included in the retrieval, respectively. Only results with AERONET AOD $\geq 0.2$ were used in the MODIS-AERONET comparison. All pixels were considered in the retrieval and each row correspond to data from 302 AERONET stations and 10354 collocated observations.

| Approximation error model | Spatial correlation model for $\tau$ and $\eta$ | $R$ | Median Bias | RMS Error |
|---|---|---|---|---|
| X | X | 0.35 | 0.14 | 0.51 |
| X | - | 0.36 | 0.16 | 0.50 |
| - | X | 0.20 | 0.11 | 1.22 |
| - | - | 0.21 | 0.11 | 1.14 |
| DT algorithm, all pixels | | 0.36 | -0.18 | 0.67 |

the best correlation between the MODIS and AERONET retrievals is observed when the approximation error model is used and spatial correlation models are turned off. These results, however, should be interpreted very carefully as they only show the global statistics. In single retrieval cases, the spatial correlation models may even play more critical role than the approximation error model. As the aerosol properties usually have clear spatial correlation we would recommend using the spatial correlation models in the retrievals.

### 5.3 Regional performance of the algorithm

The global and regional results of the DT and BDT AOD retrievals with respect to the AERONET are shown in Table 4. The results show that the BDT AOD retrievals are significantly better than the DT retrievals globally and in most of the regions. The BDT algorithm performed better than or equal to the DT algorithm in all regions when measured in RMS error, correlation



**Global**

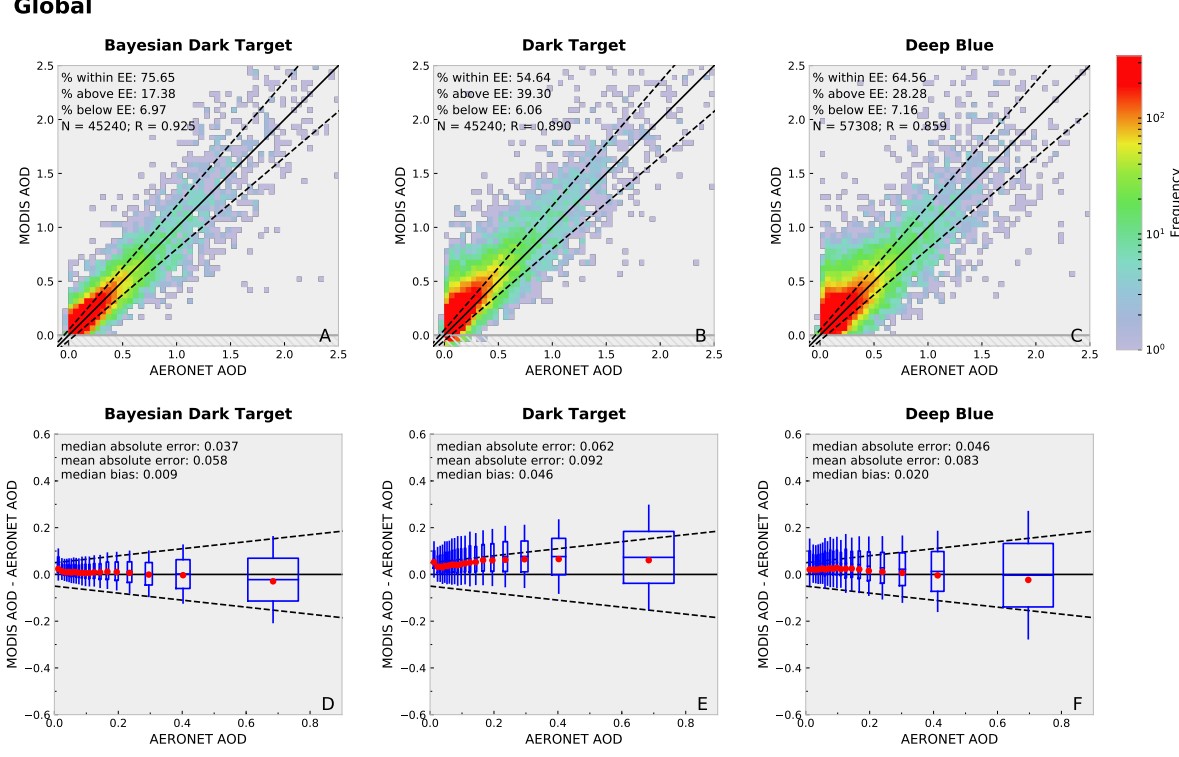

**Figure 4.** Top row: Scatter density histograms comparing global AERONET and MODIS Bayesian Dark Target (A), MODIS Dark Target (B), and MODIS Deep Blue (C) AOD retrievals. The solid black line represents the 1:1 line and the dashed lines the MODIS Dark Target expected error envelope. Bottom row: The retrieval error for MODIS Bayesian Dark Target (D), MODIS Dark Target (E), and MODIS Deep Blue (F) retrievals plotted as function of AERONET AOD. The red dots and the horizontal lines inside the boxes represent the median and mean values of MODIS AOD error, respectively. The box height and whiskers represent the 1 and 2 standard deviation intervals of the MODIS AOD retrieval error, respectively. The width of the box corresponds to the standard deviations of the AOD bin.

coefficient $R$ and fraction of retrievals inside the EE envlope. The AOD median bias is slightly worse only in Oceania (OCE; DT median bias -0.01, BDT median bias 0.02). The table shows that the largest improvements in the retrieval accuracy are seen in North America. The fraction of retrievals inside the EE envelope was increased from 57% to 81% in East North America (ENA) and from 43% to 77% in West North America (WNA) when BDT retrieval was used instead of DT. The worst regional performance when measured with the correlation with AERONET AOD is in Europe (EUR). The worst regional performance when measured with the fraction of retrievals inside the EE envelope in BDT algorithm it is in the North Africa/Middle East (NAME) region. This is probably explained by the surface type and frequent dust events in the region. It is also possible that the BDT algorithm may weight the fine aerosol model too much in this area resulting in reduced retrieval accuracy for AOD.





**Global**

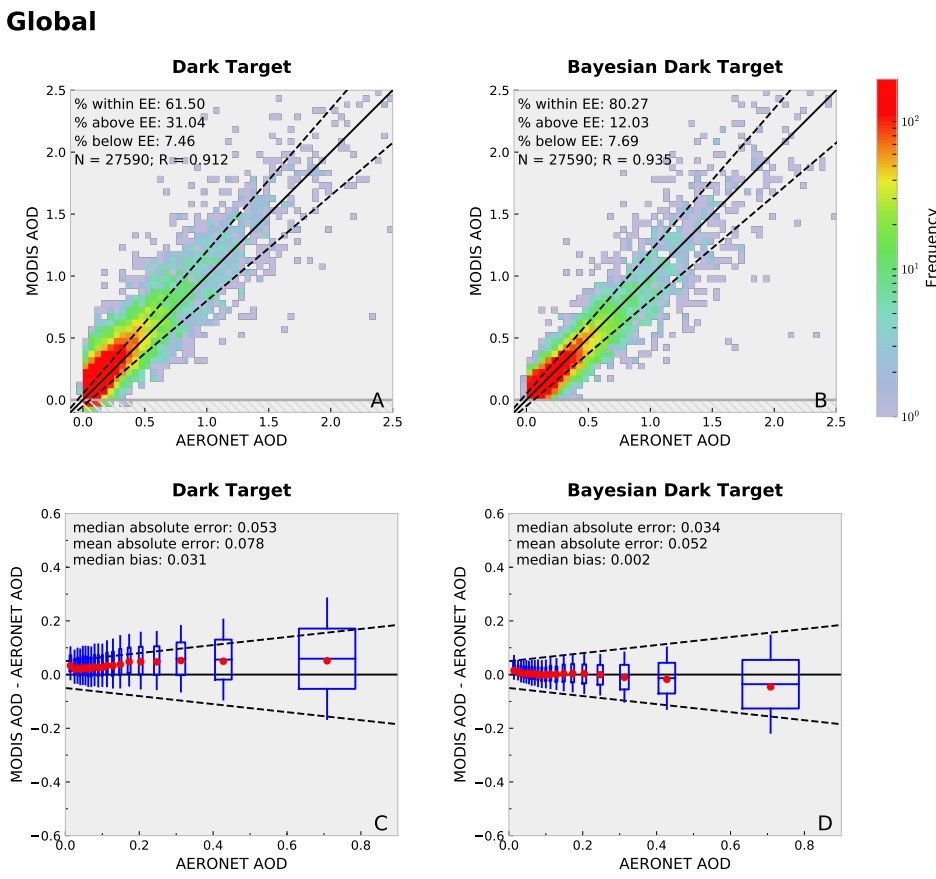

**Figure 5.** Similar figure as Figure 4 but only for MODIS Dark Target and MODIS Bayesian Dark Target algorithms and corresponding only to pixels with MODIS DT quality assurance class value of 3.

The global and regional results of the DT and BDT AE retrievals are shown in Table 5. The BDT AE retrievals have lower RMS error than the DT AE retrievals in all regions except in the Northeast Asia (NEA). The median bias in the retrieved AE is also smaller with BDT in most of the regions. In North Africa/Middle East (NAME), South Africa (SA), and Southeast Asia (SEA) the bias is, however, larger in the BDT retrievals. Especially in North Africa/Middle East region, the median bias is

5    significantly higher in BDT retrievals and this presumably is an indication of the problems in correctly retrieving the AE in dust cases over relatively bright surfaces.

Global and regional AOD accuracy comparisons between the BDT and DB retrievals are shown in Table 6. The results show that the retrieval accuracy of BDT is clearly better than the one of DB. All retrieval metrics are similar or better for BDT algorithm in all regions except in Oceania (OCE) where the DB median bias is slightly better. Figures of retrieval comparisons

10    between the BDT and DB algorithms are in the supplement.



**Table 4.** Global and regional statistics of AOD retrievals for Dark Target (DT) and Bayesian Dark Target (BDT) retrieval algorithms. All DT quality assurance classes considered. Bolded numbers indicate the algorithm with better performance.

| Region | Number of Sites DT | BDT | Number of Matches DT | BDT | $R$ DT | BDT | Median Bias DT | BDT | $f$ Within $EE_{DT}$ DT | BDT | RMS Error DT | BDT |
|---|---|---|---|---|---|---|---|---|---|---|---|---|
| Global | 346 | 346 | 45240 | 45240 | 0.89 | **0.92** | 0.05 | **0.01** | 0.55 | **0.76** | 0.14 | **0.10** |
| Global, AOD > 0.2 | 302 | 302 | 10354 | 10354 | 0.88 | **0.90** | 0.06 | **-0.00** | 0.53 | **0.68** | 0.22 | **0.17** |
| ENA | 90 | 90 | 7384 | 7384 | 0.90 | **0.94** | 0.04 | **0.01** | 0.57 | **0.81** | 0.11 | **0.06** |
| WNA | 69 | 69 | 7191 | 7191 | 0.83 | **0.92** | 0.07 | **0.01** | 0.43 | **0.77** | 0.17 | **0.09** |
| CSA | 46 | 46 | 4537 | 4537 | 0.81 | **0.89** | 0.04 | **0.00** | 0.53 | **0.76** | 0.12 | **0.07** |
| EUR | 114 | 114 | 14991 | 14991 | 0.79 | **0.83** | 0.04 | **0.01** | 0.60 | **0.79** | 0.11 | **0.06** |
| NAME | 47 | 47 | 1486 | 1486 | 0.89 | **0.91** | 0.04 | **0.00** | 0.46 | **0.57** | 0.17 | **0.15** |
| SA | 11 | 11 | 1066 | 1066 | 0.85 | **0.91** | **-0.01** | 0.01 | 0.64 | **0.75** | 0.11 | **0.09** |
| NEA | 47 | 47 | 2862 | 2862 | **0.94** | **0.94** | 0.07 | **0.00** | 0.51 | **0.71** | 0.17 | **0.12** |
| SEA | 69 | 69 | 4327 | 4327 | 0.88 | **0.89** | 0.06 | **0.01** | 0.54 | **0.61** | 0.23 | **0.18** |
| OCE | 19 | 19 | 1396 | 1396 | **0.93** | **0.93** | **-0.01** | 0.02 | 0.65 | **0.69** | 0.15 | **0.11** |

**Table 5.** Global and regional statistics of Angstrom exponent retrievals for Dark Target (DT) and Bayesian Dark Target (BDT) retrieval algorithms. All DT QA flags considered. Only retrievals with AERONET AOD larger than 0.2 were included. Bolded numbers indicate the algorithm with better performance.

| Region | Number of Sites DT | BDT | Number of Matches DT | BDT | $R$ DT | BDT | Median Bias DT | BDT | RMS Error DT | BDT |
|---|---|---|---|---|---|---|---|---|---|---|
| Global | 302 | 302 | 10354 | 10354 | **0.36** | 0.35 | -0.18 | **0.14** | 0.67 | **0.51** |
| ENA | 68 | 68 | 868 | 868 | **0.43** | 0.18 | -0.28 | **-0.08** | 0.66 | **0.50** |
| WNA | 51 | 51 | 499 | 499 | **0.24** | 0.20 | -0.34 | **-0.22** | 0.79 | **0.56** |
| CSA | 35 | 35 | 662 | 662 | 0.34 | **0.40** | -0.13 | **0.01** | 0.84 | **0.61** |
| EUR | 98 | 98 | 2679 | 2679 | 0.40 | **0.45** | -0.26 | **0.06** | 0.67 | **0.50** |
| NAME | 26 | 26 | 597 | 597 | 0.13 | **0.46** | **0.25** | 0.57 | 0.89 | **0.68** |
| SA | 10 | 10 | 425 | 425 | 0.23 | **0.40** | **0.04** | 0.16 | 1.17 | **0.34** |
| NEA | 44 | 44 | 1230 | 1230 | **0.42** | 0.13 | -0.18 | **0.17** | **0.46** | 0.57 |
| SEA | 62 | 62 | 3200 | 3200 | 0.40 | **0.44** | **-0.15** | 0.22 | 0.47 | **0.40** |
| OCE | 14 | 14 | 194 | 194 | **0.12** | -0.10 | -0.21 | **0.09** | 1.08 | **0.91** |





**Global**

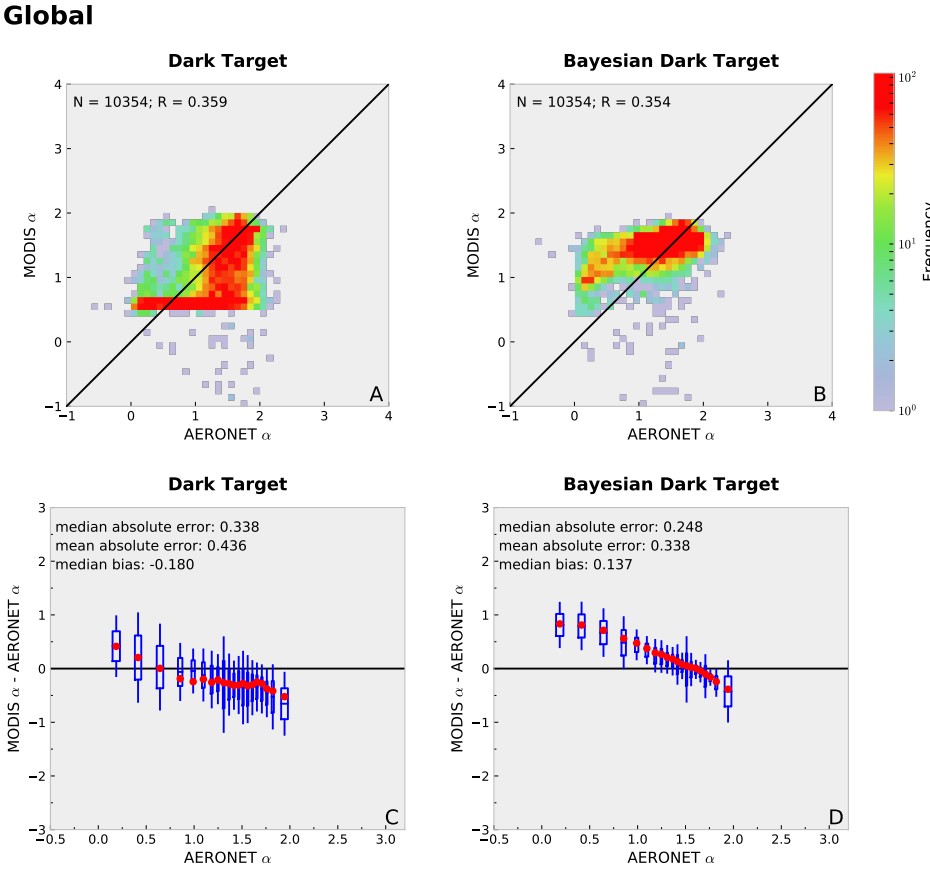

**Figure 6.** Top row: Scatter density histograms comparing global AERONET and MODIS Dark Target (A) and MODIS Bayesian Dark Target (B) Ångström exponent retrievals. The solid black line represents the 1:1 line. Bottom row: The retrieval error for MODIS Dark Target (C) and MODIS Bayesian Dark Target (D) retrievals plotted as function of AERONET Ångström exponent. The red dots and horizontal lines inside the boxes represent the median and mean values of MODIS Ångström error. The box height and whiskers represent the 1-$\sigma$ and 2-$\sigma$ intervals of the MODIS Ångström retrieval error. The width of the box corresponds to the 1-$\sigma$ of Ångström exponent bin.

## 5.4 Retrieval over urban areas

AOD retrievals over urban areas were evaluated by comparing the MODIS AOD retrievals over AERONET stations that are located in urban areas. We selected 17 AERONET stations for this comparison and the results are presented in Table 7. Results indicate that the BDT AOD retrievals are significantly better than the DT retrievals in all but one station (Mexico City). As discussed in Section 5.1, the properties of the surface reflectance in urban areas might not be well represented in the DT retrievals. The problem with urban surfaces in DT is a well-known problem and in Gupta et al. (2016b) a modified surface reflectance relationship was proposed to be used over urban areas. BDT algorithm seems to better handle the urban surfaces



**Table 6.** Global and regional statistics of AOD retrievals for Deep Blue (DB) and Bayesian Dark Target (BDT) retrieval algorithms. All pixels are considered. Bolded numbers indicate the algorithm with better performance.

| Region | Number of Sites | | Number of Matches | | $R$ | | Median Bias | | $f$ Within $EE_{DT}$ | | RMS Error | |
|---|---|---|---|---|---|---|---|---|---|---|---|---|
| | DB | BDT | DB | BDT | DB | BDT | DB | BDT | DB | BDT | DB | BDT |
| Global | 361 | 346 | 57308 | 45240 | 0.86 | **0.92** | 0.02 | **0.01** | 0.65 | **0.76** | 0.15 | **0.10** |
| Global, AOD > 0.2 | 322 | 302 | 13531 | 10354 | 0.85 | **0.90** | **0.00** | -0.00 | 0.56 | **0.68** | 0.23 | **0.17** |
| ENA | 92 | 90 | 8313 | 7384 | 0.74 | **0.94** | 0.03 | **0.01** | 0.66 | **0.81** | 0.14 | **0.06** |
| WNA | 71 | 69 | 8990 | 7191 | 0.85 | **0.92** | 0.02 | **0.01** | 0.67 | **0.77** | 0.14 | **0.09** |
| CSA | 53 | 46 | 5200 | 4537 | 0.77 | **0.89** | 0.01 | **0.00** | 0.68 | **0.76** | 0.10 | **0.07** |
| EUR | 127 | 114 | 18860 | 14991 | 0.71 | **0.83** | 0.01 | 0.01 | 0.71 | **0.79** | 0.11 | **0.06** |
| NAME | 61 | 47 | 3497 | 1486 | 0.84 | **0.91** | 0.04 | **0.00** | 0.49 | **0.57** | 0.18 | **0.15** |
| SA | 13 | 11 | 1718 | 1066 | 0.77 | **0.91** | 0.02 | **0.01** | 0.53 | **0.75** | 0.12 | **0.09** |
| NEA | 54 | 47 | 3820 | 2862 | **0.94** | **0.94** | 0.03 | **0.00** | 0.61 | **0.71** | 0.18 | **0.12** |
| SEA | 75 | 69 | 5179 | 4327 | 0.84 | **0.89** | 0.03 | **0.01** | 0.46 | **0.61** | 0.23 | **0.18** |
| OCE | 20 | 19 | 1731 | 1396 | 0.90 | **0.93** | **0.01** | 0.02 | **0.69** | **0.69** | 0.15 | **0.11** |

than the DT algorithm and carries out the AOD retrieval with similar accuracy as for the surrounding regions. Table 7 also shows the mean black sky surface albedo for the year 2015 near the AERONET station based on MCD43D3 product. There seems to be no clear connection between the black sky surface albedo and the retrieval accuracy. More detailed results from the comparison between the BDT and DB retrievals over urban areas is shown in the supplement.

## 5 5.5 Per-pixel posterior uncertainty estimates of the retrieved parameters

The BDT algorithm provides approximate posterior uncertainties for retrieved quantities. We evaluate the AOD posterior uncertainty estimates of the BDT algorithm by comparing them to the discrepancies between the BDT retrievals and AERONET observations. Table 8 shows comparison of the uncertainty estimates and the retrieval errors as a function of AERONET AOD. Credibility intervals corresponding to the MODIS DT EE envelope are also computed and presented in the table. The table

10 shows that BDT is capable of producing feasible uncertainty estimates. The comparison with the DT EE based uncertainty estimates show that the BDT pixel based uncertainties give on average more realistic estimates for the uncertainties related to the retrieved quantities over AERONET stations. On average the BDT uncertainty estimates were slightly larger than the true retrieval errors. In addition, the results also show that the BDT uncertainty estimates corresponding to large AOD values are often overoptimistic. This means that the pixel-level uncertainty estimates tend to be too low when the AOD is larger than 0.5.

## 15 6 Conclusions

A new AOD retrieval algorithm, Bayesian Dark Target (BDT), was developed. The algorithm is based on the widely used MODIS DT algorithm. In the BDT algorithm, the inverse retrieval problem is formulated in a statistical (Bayesian) framework



**Table 7.** Statistics of AOD retrievals for Dark Target (DT) and Bayesian Dark Target (BDT) retrieval algorithms over urban AERONET stations. The location information for the AERONET sites can be found at the AERONET webpage https://aeronet.gsfc.nasa.gov/.

| AERONET station | Surface albedo at 550 nm | Number of Matches | | $R$ | | Median Bias | | $f$ Within $EE_{DT}$ | | RMS Error | |
|---|---|---|---|---|---|---|---|---|---|---|---|
| | | DT | BDT | DT | BDT | DT | BDT | DT | BDT | DT | BDT |
| CCNY | 0.08 | 127 | 127 | 0.81 | **0.88** | 0.16 | **0.03** | 0.10 | **0.70** | 0.23 | **0.08** |
| Toronto | 0.09 | 189 | 189 | 0.91 | **0.96** | 0.16 | **0.02** | 0.09 | **0.74** | 0.21 | **0.08** |
| GSFC | 0.06 | 213 | 213 | 0.91 | **0.94** | 0.05 | **-0.00** | 0.59 | **0.89** | 0.09 | **0.05** |
| MD_Science_Center | 0.08 | 190 | 190 | 0.92 | **0.94** | 0.06 | **-0.02** | 0.52 | **0.82** | 0.12 | **0.06** |
| BSRN_BAO_Boulder | 0.10 | 243 | 243 | 0.85 | **0.87** | 0.08 | **0.00** | 0.35 | **0.86** | 0.10 | **0.04** |
| Univ_of_Houston | 0.10 | 134 | 134 | **0.90** | 0.83 | 0.10 | **-0.02** | 0.28 | **0.82** | 0.13 | **0.05** |
| CalTech | 0.09 | 93 | 93 | 0.60 | **0.77** | 0.11 | **-0.03** | 0.29 | **0.77** | 0.16 | **0.06** |
| El_Segundo | 0.10 | 224 | 224 | 0.37 | **0.58** | 0.38 | **0.02** | 0.00 | **0.76** | 0.45 | **0.06** |
| Mexico_City | 0.08 | 104 | 104 | **0.60** | 0.59 | **0.05** | -0.09 | **0.55** | 0.38 | 0.18 | **0.16** |
| Sao_Paulo | 0.09 | 100 | 100 | 0.69 | **0.72** | **0.04** | -0.04 | 0.69 | **0.76** | 0.09 | **0.07** |
| Paris | 0.10 | 136 | 136 | 0.75 | **0.79** | 0.07 | **0.01** | 0.43 | **0.76** | 0.13 | **0.07** |
| Thessaloniki | 0.07 | 361 | 361 | **0.89** | 0.88 | 0.04 | **0.01** | 0.68 | **0.84** | 0.09 | **0.05** |
| Moscow_MSU_MO | 0.10 | 121 | 121 | 0.83 | **0.89** | 0.06 | **-0.02** | 0.50 | **0.93** | 0.10 | **0.05** |
| Beijing-CAMS | 0.10 | 242 | 242 | **0.96** | 0.94 | 0.24 | **0.02** | 0.20 | **0.68** | 0.29 | **0.20** |
| Osaka | 0.09 | 127 | 127 | 0.74 | **0.76** | 0.17 | **0.02** | 0.18 | **0.65** | 0.24 | **0.12** |
| Kanpur | 0.11 | 254 | 254 | 0.82 | **0.91** | 0.14 | **-0.03** | 0.46 | **0.76** | 0.27 | **0.14** |
| Singapore | 0.07 | 23 | 23 | 0.95 | **0.96** | 0.38 | **0.30** | 0.04 | **0.22** | 0.75 | **0.53** |

**Table 8.** Fraction of AERONET AODs inside 50%, 80%, 90%, 95%, and 99% credible intervals based on MODIS BDT uncertainty estimates. For comparison also MODIS DT expected error (EE) envelope based results are shown corresponding to DT retrievals.

| | Fraction of AERONET AODs inside the $N\%$ credible interval based on MODIS BDT uncertainty estimates. | | | | |
|---|---|---|---|---|---|
| | $N = 50\%$ | $N = 80\%$ | $N = 90\%$ | $N = 95\%$ | $N = 99\%$ |
| MODIS DT EE based, all retrievals | 40.5 % | 68.4 % | 79.8 % | 86.7 % | 94.3 % |
| All retrievals | 59.5 % | 84.6 % | 91.4 % | 94.7 % | 97.9 % |
| 0.0 < AERONET AOD < 0.1 | 66.8 % | 89.7 % | 94.7 % | 97.0 % | 99.0 % |
| 0.1 < AERONET AOD < 0.2 | 57.4 % | 85.1 % | 92.3 % | 95.8 % | 98.6 % |
| 0.2 < AERONET AOD < 0.3 | 52.2 % | 80.3 % | 89.2 % | 93.6 % | 97.5 % |
| 0.3 < AERONET AOD < 0.5 | 46.0 % | 74.4 % | 84.3 % | 89.8 % | 95.8 % |
| 0.5 < AERONET AOD < 1.0 | 39.1 % | 65.2 % | 76.6 % | 83.6 % | 91.9 % |
| 1.0 < AERONET AOD < 2.5 | 29.2 % | 52.3 % | 64.1 % | 73.3 % | 84.0 % |
| 2.5 < AERONET AOD < 5.0 | 23.2 % | 44.8 % | 56.0 % | 64.5 % | 75.7 % |





that allows systematic use of probabilistic models for prior information and approximation errors related to inaccuracies in the physical observation models, and pixel-based uncertainty quantification for the retrieved parameters. In the BDT algorithm, the retrieved unknown parameters are the total AOD at 550 nm, FMF, and surface reflectances at 446 nm, 550nm, 644 nm, and 2.1 $\mu$m. The retrieval is carried out simultaneously in all the pixels of a granule.

The BDT algorithm was evaluated by retrieving all MODIS granules from the year 2015 and compared with AERONET AOD and AE. Results showed that by using the BDT algorithm the accuracy of the AOD retrievals was significantly improved when compared to both DT and Deep Blue (DB) retrievals. Globally, the fraction of AOD retrievals inside the DT EE envelope increased from 55% to 76% when BDT was used instead of DT. Moreover, the median bias in AOD was improved, globally the bias was 0.01 while the bias of the DT algorithm was 0.05. The AOD retrievals were improved in all studied regions and the

largest improvement was found in North America. Oceania was the region with the smallest improvement. The AE retrievals were also improved in most of the regions when BDT was used instead of the DT algorithm, however, the improvement was not as clear as for the AOD. The reason why the AE did not improve similarly as the AOD retrievals is a topic of future research.

The BDT algorithm gives approximate posterior uncertainties in the retrieved parameters for each pixel. We compared the AOD uncertainty estimates with absolute values of retrieval errors over AERONET stations. The results show that BDT is

capable of producing feasible uncertainty estimates for AOD.

The average retrieval time with the BDT algorithm was less than one minute per granule on a modern personal computer and therefore the computational costs of the algorithm allow the use of BDT for near real time processing of MODIS data. The BDT algorithm is not restricted to MODIS retrievals only and by writing the observation models for different instruments it is possible to extend the algorithm to be used for aerosol retrievals with other instruments as well. The results show that

modeling and taking into account the spatial correlations of unknown parameters and model uncertainties in the retrieval may significantly improve the accuracy of the retrievals. The inversion framework is not restricted to aerosol retrieval only and could be used for other types of remote sensing applications such as cloud and trace gas retrievals.

The first version of the BDT algorithm was constructed especially to evaluate the feasibility and accuracy of the new modeling and inversion approach and many models and selections can still be improved to make the algorithm better. The planned

improvements for the BDT algorithm in the future include:

- Use of all possible MODIS bands. BDT algorithm is capable of utilizing all possible data and use of more MODIS bands will most likely improve the retrieval accuracy.

- Spatial correlation models for the surface reflectance. More accurate models for the surface reflectance would improve the retrieval accuracy.

- Retrievals over bright surfaces. Extension of the algorithm to retrievals over bright-reflecting surfaces is a straightforward task as the Deep Blue retrievals have already shown that it is possible to use MODIS data for aerosol retrievals over bright surfaces.





- High-resolution retrievals. In high-resolution pixel-by-pixel retrievals the signal to noise ratio is low and may lead to poor retrieval accuracy. BDT takes into account the spatial correlations of aerosol properties. Therefore the use of BDT would especially improve the high-resolution (3 km) aerosol retrievals.

- Data fusion with AERONET. In the statistical inversion framework it is a straightforward task to include other data sources into the retrieval. Use of both MODIS and AERONET data together in a joint retrieval would combine the wide coverage of MODIS and the accuracy of AERONET for producing improved retrievals of the parameters.

*Code and data availability.* The MAC-v2 climatology used for prior models was downloaded from ftp://ftp-projects.zmaw.de/aerocom/climatology/MACv2_2017/550nm_2005/. The radiative transfer lookup tables that are publicly available with Dark Target standalone code at https://darktarget.gsfc.nasa.gov/reference/code were used in BDT. The AERONET V3 data used in
this study were downloaded from the NASA AERONET server at http://aeronet.gsfc.nasa.gov/. The MODIS data used in this study were downloaded from the NASA Level 1 and Atmosphere Archive and Distribution System (LAADS) at https://ladsweb.nascom.nasa.gov/. The Bayesian Dark Target algorithm code, short documentation, and prior and uncertainty models are available upon request (antti.lipponen@fmi.fi) and will be made generally available by the time of release of the final version of this paper.

## Appendix A: Derivation of the optimization problem in Equation (2)

Let

$$\boldsymbol{\rho}^{\mathrm{TOA}} = \boldsymbol{f}(\boldsymbol{\tau}, \boldsymbol{\eta}, \boldsymbol{\rho}^s; \boldsymbol{\gamma}) + \boldsymbol{e} \tag{A1}$$

be the observation model describing the relationship between the AOD $\boldsymbol{\tau}$, FMF $\boldsymbol{\eta}$, the surface reflectances at 466 nm, 552 nm, 644 nm, and 2.1 $\mu$m $\boldsymbol{\rho}^s$, the measurement geometry and aerosol model related parameters $\boldsymbol{\gamma}$ and the simulated TOA reflectances $\boldsymbol{\rho}^{\mathrm{TOA}} = \left[ \boldsymbol{\rho}^{\mathrm{TOA}}_{466\mathrm{nm}}, \boldsymbol{\rho}^{\mathrm{TOA}}_{552\mathrm{nm}}, \boldsymbol{\rho}^{\mathrm{TOA}}_{644\mathrm{nm}}, \boldsymbol{\rho}^{\mathrm{TOA}}_{2.1\mu m}, \right]$. The measurement noise and model related uncertainties are included
in the additive noise term $\boldsymbol{e}$. It should be noted that all the above variables represent the values of all dark surface pixels in a granule and are therefore vector valued. The complete statistical model of a statistical inverse problem is the posterior distribution

$$\pi\left(\boldsymbol{\tau}, \boldsymbol{\eta}, \boldsymbol{\rho}^s | \boldsymbol{\rho}^{\mathrm{TOA}}\right) \tag{A2}$$

that is the conditional joint probability distribution for AOD $\boldsymbol{\tau}$, FMF $\boldsymbol{\eta}$ and surface reflectance $\boldsymbol{\rho}^s$ values given the observed
TOA reflectances $\boldsymbol{\rho}^{\mathrm{TOA,MODIS}}$. Here $\pi$ denotes a probability distribution. From the posterior distribution different point estimates and uncertainty estimates are usually computed and used to infer the retrieved parameters.

Applying the well-known Bayes' theorem to the posterior distribution (A2) it can be written as

$$\begin{aligned} \pi\left(\boldsymbol{\tau}, \boldsymbol{\eta}, \boldsymbol{\rho}^s | \boldsymbol{\rho}^{\mathrm{TOA}}\right) &= \frac{\pi\left(\boldsymbol{\rho}^{\mathrm{TOA}} | \boldsymbol{\tau}, \boldsymbol{\eta}, \boldsymbol{\rho}^s\right) \pi\left(\boldsymbol{\tau}, \boldsymbol{\eta}, \boldsymbol{\rho}^s\right)}{\pi\left(\boldsymbol{\rho}^{\mathrm{TOA}}\right)} \\ &\propto \pi\left(\boldsymbol{\rho}^{\mathrm{TOA}} | \boldsymbol{\tau}, \boldsymbol{\eta}, \boldsymbol{\rho}^s\right) \pi\left(\boldsymbol{\tau}, \boldsymbol{\eta}, \boldsymbol{\rho}^s\right) \end{aligned} \tag{A3}$$



where $\pi\left(\boldsymbol{\rho}^{\mathrm{TOA}}|\boldsymbol{\tau},\boldsymbol{\eta},\boldsymbol{\rho}^s\right)$ is the likelihood distribution describing the relationship between the observed reflectances and the unknown parameters, $\pi\left(\boldsymbol{\tau},\boldsymbol{\eta},\boldsymbol{\rho}^s\right)$ is the prior distribution that can be used to model the information we have on unknown parameters (e.g. non-negativity, spatial correlation structures etc.) prior to the observations, and $\pi\left(\boldsymbol{\rho}^{\mathrm{TOA}}\right)$ is the evidence term that describes the probability of the event we observe. Usually the evidence term is unknown but as the observations have

already been made and the value of $\boldsymbol{\rho}^{\mathrm{TOA}}$ is fixed it may be treated as a normalization constant which is not needed in the computation of the estimates.

We model the AOD, FMF, and surface reflections as uncorrelated and the noise term $\boldsymbol{e}$, as an additive noise observation model (A1). Thus the likelihood distribution takes the form (Kolehmainen et al., 2011; Kaipio and Kolehmainen, 2013)

$$\pi\left(\boldsymbol{\rho}^{\mathrm{TOA}}|\boldsymbol{\tau},\boldsymbol{\eta},\boldsymbol{\rho}^s\right) = \pi_e\left(\boldsymbol{\rho}^{\mathrm{TOA,MODIS}} - f(\boldsymbol{\tau},\boldsymbol{\eta},\boldsymbol{\rho}^s;\boldsymbol{\gamma})\right) \tag{A4}$$

where $\pi_e$ is the probability distribution of the measurement noise and approximation errors $\boldsymbol{e}$, and $\boldsymbol{\rho}^{\mathrm{TOA,MODIS}}$ denotes the actual reflectances measured by the MODIS.

Combining (A3) and (A4) we get that

$$\pi\left(\boldsymbol{\tau},\boldsymbol{\eta},\boldsymbol{\rho}^s|\boldsymbol{\rho}^{\mathrm{TOA}}\right) \propto \pi_e\left(\boldsymbol{\rho}^{\mathrm{TOA,MODIS}} - \boldsymbol{f}(\boldsymbol{\tau},\boldsymbol{\eta},\boldsymbol{\rho}^s;\boldsymbol{\gamma})\right)\pi\left(\boldsymbol{\tau},\boldsymbol{\eta},\boldsymbol{\rho}^s\right). \tag{A5}$$

In this study, we model the term $\boldsymbol{e}$ as a Gaussian distributed random variable:

$$\boldsymbol{e} \sim \mathcal{N}\left(\mathbb{E}_e,\boldsymbol{\Gamma}_e\right) \tag{A6}$$

where $\mathbb{E}_e$ and $\boldsymbol{\Gamma}_e$ denote the expected value and covariance matrix of $\boldsymbol{e}$. We also model the prior information for AOD $\boldsymbol{\tau}$, FMF $\boldsymbol{\eta}$, and surface reflectance $\boldsymbol{\rho}^s$ as Gaussian distributed. Furthermore, we assume that AOD $\boldsymbol{\tau}$, FMF $\boldsymbol{\eta}$ and surface reflectances $\boldsymbol{\rho}^s$ are mutually uncorrelated with prior models:

$$\boldsymbol{\tau} \quad \sim \quad \mathcal{N}\left(\mathbb{E}_{\boldsymbol{\tau}},\boldsymbol{\Gamma}_{\boldsymbol{\tau}}\right) \tag{A7}$$

$$\boldsymbol{\eta} \quad \sim \quad \mathcal{N}\left(\mathbb{E}_{\boldsymbol{\eta}},\boldsymbol{\Gamma}_{\boldsymbol{\eta}}\right) \tag{A8}$$

$$\boldsymbol{\rho}^s \quad \sim \quad \mathcal{N}\left(\mathbb{E}_{\boldsymbol{\rho}^s},\boldsymbol{\Gamma}_{\boldsymbol{\rho}^s}\right) \tag{A9}$$

$$\pi(\boldsymbol{\tau},\boldsymbol{\eta},\boldsymbol{\rho}^s) \quad = \quad \pi(\boldsymbol{\tau})\pi(\boldsymbol{\eta})\pi(\boldsymbol{\rho}^s). \tag{A10}$$

These selections result in a posterior distribution

$$
\begin{aligned}
\pi\left(\boldsymbol{\tau},\boldsymbol{\eta},\boldsymbol{\rho}^s|\boldsymbol{\rho}^{\mathrm{TOA}}\right) \quad \propto \quad & \pi_e\left(\boldsymbol{\rho}^{\mathrm{TOA,MODIS}} - \boldsymbol{f}(\boldsymbol{\tau},\boldsymbol{\eta},\boldsymbol{\rho}^s;\boldsymbol{\gamma})\right)\pi\left(\boldsymbol{\tau}\right)\pi\left(\boldsymbol{\eta}\right)\pi\left(\boldsymbol{\rho}^s\right) \\
\propto \quad & \exp\left\{-\frac{1}{2}(\boldsymbol{\rho}^{\mathrm{TOA,MODIS}} - \boldsymbol{f}(\boldsymbol{\tau},\boldsymbol{\eta},\boldsymbol{\rho}^s;\boldsymbol{\gamma}) - \mathbb{E}_e)^{\mathrm{T}}\boldsymbol{\Gamma}_e^{-1}(\boldsymbol{\rho}^{\mathrm{TOA,MODIS}} - \boldsymbol{f}(\boldsymbol{\tau},\boldsymbol{\eta},\boldsymbol{\rho}^s;\boldsymbol{\gamma}) - \mathbb{E}_e)\right. \\
& \left. -\frac{1}{2}(\boldsymbol{\tau} - \mathbb{E}_{\boldsymbol{\tau}})^{\mathrm{T}}\boldsymbol{\Gamma}_{\boldsymbol{\tau}}^{-1}(\boldsymbol{\tau} - \mathbb{E}_{\boldsymbol{\tau}}) - \frac{1}{2}(\boldsymbol{\eta} - \mathbb{E}_{\boldsymbol{\eta}})^{\mathrm{T}}\boldsymbol{\Gamma}_{\boldsymbol{\eta}}^{-1}(\boldsymbol{\eta} - \mathbb{E}_{\boldsymbol{\eta}}) - \frac{1}{2}(\boldsymbol{\rho}^s - \mathbb{E}_{\boldsymbol{\rho}^s})^{\mathrm{T}}\boldsymbol{\Gamma}_{\boldsymbol{\rho}^s}^{-1}(\boldsymbol{\rho}^s - \mathbb{E}_{\boldsymbol{\rho}^s})\right\}.
\end{aligned} \tag{A11}
$$

We select to look for the parameters $\boldsymbol{\tau},\boldsymbol{\eta},\boldsymbol{\rho}^s$ that maximize the value of the posterior distribution (A11). This estimate is known as the *maximum a posteriori* (MAP) estimate. The MAP estimate may be found at the minimum of the minus logarithm of the





posterior distribution as:

$$(\boldsymbol{\tau}, \boldsymbol{\eta}, \boldsymbol{\rho}^s)_{\mathrm{MAP}} = \arg\min_{\boldsymbol{\tau}, \boldsymbol{\eta}, \boldsymbol{\rho}^s} \left( \left\| L_e \left( \boldsymbol{\rho}^{\mathrm{TOA,MODIS}} - \boldsymbol{f}(\boldsymbol{\tau}, \boldsymbol{\eta}, \boldsymbol{\rho}^s; \boldsymbol{\gamma}) - \mathbb{E}_e \right) \right\|^2 + \left\| \mathbf{L}_{\boldsymbol{\tau}} \left( \boldsymbol{\tau} - \mathbb{E}_{\boldsymbol{\tau}} \right) \right\|^2 + \left\| \mathbf{L}_{\boldsymbol{\eta}} \left( \boldsymbol{\eta} - \mathbb{E}_{\boldsymbol{\eta}} \right) \right\|^2 + \left\| \mathbf{L}_{\boldsymbol{\rho}^s} \left( \boldsymbol{\rho}^s - \mathbb{E}_{\boldsymbol{\rho}^s} \right) \right\|^2 \right)$$

(A12)

where $\mathbf{L}_e$, $\mathbf{L}_{\boldsymbol{\tau}}$, $\mathbf{L}_{\boldsymbol{\eta}}$ and $\mathbf{L}_{\boldsymbol{\rho}^s}$ are the Cholesky factors of the inverse covariance matrices $\boldsymbol{\Gamma}_e^{-1}$, $\boldsymbol{\Gamma}_{\boldsymbol{\tau}}^{-1}$, $\boldsymbol{\Gamma}_{\boldsymbol{\eta}}^{-1}$ and $\boldsymbol{\Gamma}_{\boldsymbol{\rho}^s}^{-1}$, respectively.

### Appendix B: Construction of the approximation error model

In the BDT algorithm, we construct an approximation error model that describes the uncertainties and inaccuracies in the simulated TOA reflectances due to imperfect models and unknown aerosol and surface parameters. The construction of the model is based on simulated TOA reflectances that are compared with the reflectances measured by the MODIS instrument. In the construction of the approximation error model, the MODIS measurements are considered as the ground truth measurements.

Let

$$\boldsymbol{\rho}^{\mathrm{TOA,simulated}} = \boldsymbol{\rho}^{\mathrm{TOA,simulated}}(\boldsymbol{\tau}, \boldsymbol{\eta}, \boldsymbol{\rho}^s; \boldsymbol{\gamma})$$

(B1)

be the TOA reflectances simulated with the DT LUT model. Here $\boldsymbol{\gamma}$ denotes the auxiliary (fixed) model parameters such as measurement geometry, surface elevation, and aerosol models. We assume that AERONET can accurately measure the aerosol properties AOD $\boldsymbol{\tau}_{\mathrm{AERONET}}$ and FMF $\boldsymbol{\eta}_{\mathrm{AERONET}}$, and that the MODIS MCD43C3 product can be used to derive accurate estimates for the surface reflectances $\boldsymbol{\rho}^{\mathrm{s,MCD43C3}}$. The contribution of uncertainties (mostly due to fixed DT aerosol and LUT

models) in the simulated TOA reflectances corresponding to a single MODIS-AERONET collocated measurement can be computed as the discrepancy between the simulated and observed TOA reflectances at the AERONET station location as:

$$\boldsymbol{n} = \boldsymbol{\rho}^{\mathrm{TOA,MODIS}} - \boldsymbol{\rho}^{\mathrm{TOA,simulated}}(\boldsymbol{\tau}_{\mathrm{AERONET}}, \boldsymbol{\eta}_{\mathrm{AERONET}}, \boldsymbol{\rho}^{\mathrm{s,MCD43C3}}; \boldsymbol{\gamma}).$$

(B2)

In the BDT algorithm, we use $N$ AERONET-MODIS collocations to compute a database of simulation-MODIS TOA reflectance discrepancies $\{n_i\}_{i=0}^N$ for all regions (shown in Figure 1) and months. We model $n$ as Gaussian multivariate random

variable and estimate the expected value and covariance matrix as sample average and sample covariance of $\{n\}$. To minimize the effect of outliers to the uncertainty model we use median instead of sample average. AERONET data does not directly include AERONET FMF $\eta_{\mathrm{AERONET}}$. Therefore, the AERONET FMF is computed using a search approach in which the FMF values $0.0, 0.05, 0.1, \ldots, 0.95, 1.0$ are tested and the one that produces the best match with the AERONET AE is selected as the $\eta_{\mathrm{AERONET}}$.

*Competing interests.* The authors declare that they have no conflict of interest.



*Acknowledgements.* We thank the AERONET PIs and their staff for establishing and maintaining the AERONET sites used in this investigation. We thank NASA MODIS team to kindly make the MODIS data publicly available. V. Kolehmainen acknowledges the Academy of Finland (Project 250215, Finnish Centre of Excellence in Inverse Problems Research).



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
