# Peer review of "Bayesian Dark Target Algorithm for MODIS AOD retrieval over land"

_Atmospheric Measurement Techniques, 2017_

## Short Comment (SC1) · 3 Nov 2017

This is a very interesting and important study which provides what appears to be a better-performing way of retrieving AOD from MODIS measurements over land than the land Dark Target (DT) and Deep Blue (DB) algorithms. I have talked with the authors a bit about their approach at recent meetings, and am glad to see a paper on the subject appear now. After a careful reading I had a few comments/questions which I was hoping the authors could expand upon.

The authors present their work as a Bayesian DT (BDT) approach, which essentially implies recasting the DT algorithm within a more formal error propagation system. As part of this statistical formalism, they also simultaneously retrieve all valid L2 pixels

in a granule, which allows the use of spatial variability constraints, rather than using the independent pixel approximation, and transform much of the data into log space to avoid unphysical negative values. This is all good stuff. I think that the manuscript is written and presented well, the approach has technical merit, and the authors appreciate some nuances about DT that others often do not (e.g. the FMF is not "fraction of AOD from the fine mode" as it is in some other data sets, but "weight of the fine-mode dominated aerosol optical model").

Digging down, there are two other major changes: (1) the 550 nm band is also used in the retrieval (DT does not use this band) and (2) surface reflectance becomes a retrieved quantity (using the MODIS BRDF/albedo product as a prior constraint) rather than the spectral shape being an assumed quantity. These both have bigger implications, and are what I have questions about.

On (1), since the authors are adding this band, they must be generating new LUTs (since there is no pre-existing DT 550 nm land LUT). I may have missed it but did not see which radiative transfer code is used to generate the LUT? Is this the same as is used in the MODIS DT algorithm? And why was the 550 nm band additionally added; what happens if it is not used, is performance comparable? I know that MODIS DT and some other algorithms choose not to use this band for retrievals over land, as assumed spectral/directional surface reflectance relationships don't always work so well for 550 nm as some other wavelengths.

Point (2) is the bigger thing. For me, the defining characteristic of the DT algorithm is the assumption that the swIR region can be used to model reflectance in the blue and red bands, according to the relationships developed first by Kaufman and then expanded by Levy. All algorithms must make some simplifying constraining assumption about surface reflectance and this is the core of what DT is and what differentiates it from other approaches. For (almost) any sensor, when the AOD is low, the dominant over-land source of AOD retrieval error comes from surface model error (since most of the signal is surface reflection), so a retrieval's surface reflectance model is the firstorder determinant of how the retrieval will behave and when it will and won't work well.

The BDT approach, on the other hand, retrieves surface reflectance simultaneously with AOD and FMF, using an aggregation of the MODIS BRDF product (which is itself a time-aggregated product based on atmospheric correction of MODIS imagery) and variability constraints to provide an a priori. In this sense these a priori constraints are the new surface model at the core of the algorithm and I expect the key to why it appears to work better than standard DT and DB. This is a bit more similar to e.g. the Deep Blue approach over deserts (to oversimplify, a climatology of surface reflectance obtained from the clearest 15% of scenes) or the MAIAC approach (where a time series of a number of days is built up and then surface and atmosphere are retrieved together) than it is to DT. The BDT algorithm has, unless I have misunderstood, entirely abandoned the swIR-to-visible surface model at the core of DT. All that appears to be in common are the aerosol optical models and cloud screening. This is not a criticism of the method, which appears sound. But it leads me to my main question: the BDT approach is clearly an approach which works well, but is it really correct to call it "Bayesian Dark Target", when the core feature of DT is the aspect which was discarded?

In my mind, it is not and it would be better to pick a different name as BDT could be misleading. The name DT conjures up the MODIS DT algorithm, and BDT likewise implies that. This is, for the reasons discussed above, something different.

On an unrelated note, Equation 3 defines the posterior covariance matrix for the retrieved state. This can be used to provide pixel-level uncertainty estimates for retrieved AOD (and other quantities), a topic of much current interest. It would be interesting to compare these to the actual AOD retrieval errors against AERONET, in a statistical sense, to assess whether these are reasonable. For example, for the subset of matchups with an actual retrieval absolute error of X, is the distribution of estimated uncertainties consistent with an expectation of an error of X? (See section 3.3 of Popp et al. 2016, doi:10.3390/rs8050421 for some other example analyses looking at validating

pixel-level uncertainties.) If yes, great. If not, when and where there is a mismatch between typical estimated uncertainties and typical actual errors can tell you something about which terms in your error budget are not quite right.

I also had a comment on the results shown in Figure 6. The high bias in Ångström exponent (AE) in both DT and BDT when the AERONET AE is low (i.e. likely cases dominated by dust) may well be because the 'coarse-dominated' aerosol model used in the retrievals assumes spherical particles, which do not model the scattering/absorption of nonspherical dust particles well. This means that the phase function is simulated poorly at some angles, and the spectral dependence of absorption and extinction is incorrect. Positive AE biases are one characteristic signature of this problem. Some theoretical simulations of this are shown in Mischenko et al 1997, doi:10.1029/96JD02110; more recently, we gave (over ocean) a practical demonstration of the differences between spherical and spheroid assumptions in Lee et al 2017, doi:10.1002/2017JD027258.

---

## Referee Comment (RC1) · A.C. Povey (Referee) · 15 Nov 2017

This paper introduces an algorithm for the retrieval of aerosol optical depth and fine mode fraction from MODIS observations ( $\tau$  and  $\eta$ , respectively). It utilises the same look-up tables, aerosol typing, and quality control as the Dark Target algorithm but (as pointed out in Dr. Sayer's comment) uses a different treatment of the surface. The most pronounced change is that it simultaneously retrieves the properties for multiple pixels, using a priori assumptions on the spatial correlation of  $\tau$  and  $\eta$  alongside a characterisation of the observational and forward model errors to better constrain the retrieval problem. The algorithm is evaluated against the current Dark Target and Deep Blue algorithms by comparison to AERONET observations during 2015. The new algorithm

estimates.

I strongly recommend the publication of this paper. The presentation is clear and concise while the validation demonstrates that it is a compelling new technique to measure aerosol properties. Below, I include some thoughts that may benefit from consideration. There, P1L2 means line 2 of page 1. (I have declined to remain anonymous as, after reading the paper, I discussed it with the authors at a recent conference which addressed many of my initial concerns.)

- P4L21 The retrieval of  $\log x$  rather than x for positive variables is well documented. You retrieve  $\log(1+x)$ , which I have not encountered before. Could you further discuss this choice, maybe providing references to demonstrate it's use elsewhere? I am concerned that it permits  $-1 \le x$ . Did you specifically wish to retain the small but negative  $\tau$  from the original Dark Target algorithm or are the 'constraints that exclude non-physical solutions' (P5L23) hard limits that prevent this behaviour? If the latter, why not use the more common  $\log x$  formulation? Have you considered how hard limits distort Gaussian uncertainty estimates near those limits?
  - §2 It's unclear from the text precisely how many pixels are processed at once. Is it an entire granule? Processing 50,000 pixels at once would be an impressive computational task!

I also recall that you only process pixels for which a DT retrieval was produced (implicitly adopting their cloud flag), but I don't find that mentioned in this text.

- P6L17 Though the 50 km correlation length is widely used, you should cite something. doi:10.1175/1520-0469(2003)060<0119:MVOTA>2.0.CO;2 is quite common.
- §3.1.3 This method contains a few surprising features. Why use blue sky albedo? Why seasonal averages? Why average the 3 closest values rather than do a bilinear or triangular interpolation? Was the technique overly sensitive to these choices

(i.e. were these chosen at random and worked or did it take several attempts to find a stable solution)?

- P8L7 What motivated the addition of coarse mode aerosol to the continental mode? Are the Dark Target team considering removing this step from their own processing?
- P9L2-4 I don't understand what you mean by 'marginalize the posterior model'. Marginalize means 'to treat as insignificant' and you use posterior model to describe the cost function, (2). I would expect one to 'minimize the posterior model', but I fail to see why that is relevant to the approximation error approach.
- P9L31 'Physical' may be a better word than 'true' here as there arguably is a 'true' FMF as defined by the Dark Target algorithm, but the point is that that value doesn't always mean something in reality.
- Fig. 1 Could the urban sites (discussed in §5.4) be displayed in a different colour?
- Figs. 2&3 For Angstrom exponent, could you use a colour bar that has grey at the centre so we can distinguish missing data from a value of 1?
  - App. B I broadly like this idea, and do something similar myself (though not yet in a published paper), but I'm curious about assuming the MODIS BRDF is accurate. It's a good retrieval but not without substantial uncertainty (of many forms - representational, approximation, etc.). Considering the dominance of the surface in aerosol error budgets, how accurate do you think these estimates of the approximation error are?
    - Several references list a URL twice. Perhaps replace the BibTeX field url with doi?

I also include some proofreading recommendations.

**C3**

- P1L21 Hyphenate satellite-based.
- P1L22 provides a means
- P2L2 the oldest still operating
- P2L4 An One algorithm to retrieve the aerosol properties, such as aerosol optical depth (AOD), is the Dark Target
- P2L7 Comma after effect.
- P2L19 be downloaded for example from
- P2L21 MODIS is the Deep Blue
- P2L26 Commas after useful and example.
- P2L31-32 Hyphenate pixel-by-pixel.
  - P3L11 Comma after information.
  - P6L4 AOD values in a 1 degree
- P6L17-18 correlation but on the other hand to allow while allowing for certain features
  - P6L18 Commas after features and plumes.
  - P8L2 we use a similar
  - P7L11 processing of a large
  - P7L19 average of the 3 closest
  - P8L23 In the statistical ... model and take into account the uncertainties

- P9L4-5 Use \citet rather than \citep for this reference to get rid of the brackets.
  - P9L9 Comma after uncorrelated.
- P9L12 Hyphenate near-real-time.
- P10L3 Hyphenate AERONET-based and size-distribution-related.
- P10L9 Comma after the brackets.
- P11L23 carried out is are selected differently in these algorithms. The DB
- P12L6 Hyphenate all-pixel. Comma after reduced.
- P12L7 the DT retrievals also in this case. This
- P15L8 Comma after area.
- P18L4 DT retrievals in at all but
- P21L17 Hyphenate near-real-time.
- P21L22 Comma after applications.
- P22L1 Comma after retrievals. Hyphenate signal-to-noise.
- P22L2 Comma after therefore.
- P22L25 Comma after posterior distribution.
- P23L4 Commas after usually and but.
- P23L5 Comma after fixed.
- P23L8 Comma after thus.

**C5**

P24L2 The first L should be bold not italic.

P24L21 we use the median

---

## Referee Comment (RC2) · Anonymous Referee #3 · 22 Nov 2017

This paper provides a detailed description and validation of a new algorithm for the retrieval of aerosol optical depth (AOD) and Fine Mode Fraction (FMF) (plus surface reflectance in four MODIS bands, although these values aren't really discussed or evaluated by the authors). The new algorithm, refereed to as the Bayesian Dark Target (BDT) Algorithm, uses the same preprocessing, look-up tables and aerosol typing as the operational MODIS Dark Target (DT) algorithm, but otherwise takes a completely different approach to the retrieval problem:

- The spectral constraints on surface reflectance employed by DT are done away with.

- Spatial correlation in AOD and FMF are used to provide a priori constraint

[Figure]

- An entire granule of MODIS observations are analysed simultaneously using a Bayesian maximum a posteriori optimisation, which provides pixel level uncertainties based on prior assumptions of measurement and forward modelling uncertainty.

The algorithm is evaluated against both the DT and the Deep Blue algorithms (both of which are operational MODIS products), using AERONET as a reference, over the year 2015. The BDT algorithm shows a significant superiority to the two operational products across almost all regions, with the authors particularly emphasising improvements over urban areas.

I am happy to recommend this paper for publication, subject to the authors addressing the following questions and corrections. In particular, I am keen to see my first two general points addressed satisfactorily. The paper is generally clearly presented and concise, and the work represents a novel and valuable addition to the field of aerosol remote sensing from satellite.

I have divided my comments and questions into general points, followed by specific points denoted by page (P) and line number (L).

**General points**

I am somewhat concerned by the way the approximation error (§3.3 and Appendix B) is computed. In §3.2 the authors state that the mean of the measurement noise PDF ($\mathbb{E}_n$) is assumed to be zero. However, it appears that this constraint is not applied when computing the approximation error mean ($\mathbb{E}_u$) from comparing MODIS TOA reflectance to values simulated from AERONET AODs. Is this correct? If it is, then this approach is making an implicit bias correction to the MODIS L1B reflectances, based on AERONET aerosol measurements and the retrieval's own forward model (plus the MCD43C surface reflectance) - this is fine as it stands, although it's clearly a bit of fudge. However, the authors then use the same AERONET measurements as validation data. It is thus not a huge surprise that they see a significant improvement

in the bias against AERONET compared to the DT and Deep Blue products. Indeed, as the correction is computed separately for different regions (the same ones used in the validation?) and seasons, we might expect it to improve the correlation and RMS of global and yearly comparisons of AOD or FMF vs AERONET as well.

If I am correct in my reading of how the approximation error is calculated and applied, then I would like to see the authors provide a comparison against AERONET where $\mathbb{E}_u$ is assumed to be zero. Otherwise, a clarification of how the approximation error is calculated is needed.

The result resented in §5.2, that the retrieval performs best vs AERONET if the prior constraint of the spatial correlation of AOD and FMF are switched off, doesn't seem make sense without considering the above point, as you are then retrieving 6 parameters (four surface reflectances + AOD + FMF) from 4 measurements. Thus the results in tables 3 and 4 need further explanation.

I think that the fact that the best correlation, bias, fraction within $EE_{DT}$ and RMS error all correspond to the configuration where spatial correlation is disabled, but approximation error is enabled, is simply due to the regional/seasonal bias correction against AERONET implicitly applied by the approximation error methodology. Again, there is nothing inherently wrong with doing such a correction, but you cannot then pretend that AERONET is an independent source of validation data.

Furthermore, tables 3 and 4 show that applying the correlation constraints don't actually improve the results vs AERONET, even if the approximation error is disabled. This would seem to imply that the correlation constraints aren't improving the retrievals at all. I would be surprised if it turned out that these constraints don't actually improve the retrieval in many cases, but I feel that a concrete example is needed, rather than the vague assurances given at the end of §5.2.

Also, am I right in thinking that if both the approximation and spatial correlation errors are switched off, the retrieval effectively assumes the prior surface reflectance values from the MCD43C product are correct (i.e. the retrieval doesn't move from the a priori

values)?

I agree with the point made in an earlier comment by Dr Sayer regarding the name of the algorithm. Clearly, given that the processing done by the authors shares the cloud-clearing, look-up tables (and thus aerosol models) etc as the DT algorithm, it would seem fair enough to call the resulting *product* as the Bayesian Dark Target product, but the retrieval algorithm itself could be described as opposite in approach to the NASA DT algorithm, as BDT does away with both the independent pixel and spectrally correlated surface reflectance assumptions which form the basis of the DT (and Deep Blue) approaches (while introducing its own assumptions about spatial correlations in aerosol properties). My feeling is that the authors are in danger of "under selling" the algorithm, as, without going through the details of the algorithm, it could appear to just be DT with pixel-by-pixel uncertainty estimates.

In a similar fashion, I'd like to see the authors move away from the (in my opinion misleading) use of the term fine-mode fraction to describe the fraction of the AOD due to the fine mode. Perhaps "fine mode AOD fraction" is a better term? At the very least, the definition of what is actually meant by FMF needs to be stated up-front.

Finally, though I appreciate that adding in an additional product into their validation would be a bit much, the authors don't seem to be aware of the MAIAC MODIS product, which (as Dr Sayer noted) in some ways has more in common with their approach than DT.

**Specific points**
P1L1: Suggest you reword to "... (BDT) algorithm for the retrieval of aerosol optical depth over land from MOderate Resolution..."

P1L18: Reword to "...particles may be hazardous to human health when inhaled..."

P1L21: Add a comma after "predictions".

P2L4: I'd describe DT as the primary operational algorithm used to retrieval aerosol, not just "an algorithm".

P2L7: Reword to "...DT algorithm is the brightening effect, whereby an increased amount of aerosol over dark..."

P2L19: Remove "for example".

P2L24-L27: The sentences describing the Deep Blue algorithm don't scan well at all.

P2L34: Please provide a reference/justification for the statement that taking advantage of spatial correlations in aerosol properties can improve retrieval results. I note that your own results (Section 5.2, Tables 3 and 4) don't actual support this statement (although I'd be surprised if it wasn't true!)

P4L8: Why did you choose the listed bands in particular? Was just because these were the ones for which look-up tables were available?

P4L21: As with the other referee, Dr Povey, I am also confused by the $\tau = \log(\tilde{\tau} + 1)$. I understand retrieving AOD in log space (not only do you avoid the problem of negative AODs, but the observed distribution of AODs is much closer to log-normal than normal), but where does the "+1" come from? Also, as Dr Povey notes, $\log(\tau + 1)$ is defined for $\tau > -1$, and thus doesn't prevent negative optical-depth values as you state in the text.

P4L22: The retrieval using log reflectance is also new to me. Can you elaborate on why this was chosen? In particular, doesn't this complicate the use of the estimates of the measurement noise of the MODIS reflectances? And, again the "+1" needs explaining.

P4L25: Please explicitly define the symbols used for expected value vectors and covariance matrices here.

P5L6: Eq (2) needs reformatting so that it fits within the margins.

P6 (§3.1.1 and 3.1.2): I don't agree with your approach in presenting the Prior models and their covariance matrices in particular. For a start, the values for $\sigma^2_{\text{nugget}}$ make up

a small fraction of the total variance (3rd sig. fig. for AOD and 2nd sig. fig. for FMF) and thus are largely irrelevant. Furthermore, I think it be clearer if you were define a correlation matrix and variance matrix separately, which are then combined to give you your covariance. The off-diagonal elements of a covariance matrix are combination of the variance of the variables concerned (where an increased value corresponds to a less tight constraint) and correlation between them (where a increased value implies a tighter constraint).

P6L16-18: The authors need to acknowledge that an inherent weakness of their scheme, and the use of a spatial correlation constraint in particular, is that it will always result in the smoothing of sharp features in the aerosol field (such as smoke plumes). There is no way you will be able to "correctly" retrieve the AOD of a near-source aerosol plume when assuming a correlation in aerosol AOD and FMF over 50 km.

P7L14: Please provide some reasoning behind the choice of the blue sky albedo, and the 0.5 weighting coefficient in particular.

P7L29: I know Eq (5) is fairly ubiquitous in aerosol and cloud remote sensing, but a reference for its derivation would be nice.

P8L11: How are the look-up tables corrected for surface elevation? Is this an additional parameter in the table?

P9L8-9: Change to "We model the approximation error u as spatially, but not spectrally, uncorrelated, meaning the correlations between MODIS bands are taken into account".

P9L10: Delete the "the" before "spatial and season variations", Also, place a comma after "Similarly".

P9L19: Remove "the" before "$\pm 30$".

P9L25: Change to "In order to evaluate the near real time..."

P10L14: Remove "the" before "DB retrievals".

P11L13: The use of "oversmoothed" here irks me somewhat. For a start, how does one define what is "oversmoothed"? I feel the authors should be up-front and acknowledge that there is some evidence in Fig. 3 that their algorithm is smoothing the AOD field to a certain degree - in particular, I am looking at the thin smoke plume

slightly above the smoother, wider area of smoke in the middle of the image. There is a clear linear feature in both the true-colour and DT images, which largely absent from the BDT results.

P11L14-17: The authors present some circumstantial arguments to suggest that the BDT FMF in Fig. 3 is superior to that provided by DT, and I agree that some of the features present in the DT results are almost certainly artefacts. However, due to the spatial constraints applied to the BDT algorithm, it is always going to produce a more pleasing looking, smoothly varying field than an independent pixel approach like DT. This doesn't mean that we can conclude that, for the scene in question, we can say which product is quantitatively more accurately. Again, some acknowledgement of the fact that some real features might be smoothed-out in the BDT product should be included.

P12L2: Insert a comma after "algorithms".

P12L4: Insert a comma after "Figure 4".

P12L8: Change to "...was carried out, based on the DT algorithm QA flag, which is designed to discard..."

P13L6: Change to "Visual inspection shows the BDT retrievals..."

P21L28: I have never heard of anyone trying to use spatial correlation constraints on the surface land reflectance. This doesn't seem to make sense as the land surface is generally pretty anisotropic at all spatial scales. Do the authors mean spectral correlation instead?

P22L1: I don't believe the claim that the signal-to-noise ratio of MODIS reflectances is too low to allow accurate aerosol retrievals. The reasons that aerosol products have usually been done on a courser spatial scale than native instrument resolution are all to do with dealing with residual cloud contamination and surface anisotropy (and reducing product size and computational cost).

P22L4: Due to the inclusion of the AERONET based approximation error term, the scheme already has a kind of data-fusion with AERONET!

P21/P22: I have an additional thought for future development of the algorithm: Is there

any reason the approach couldn't be applied over the ocean?

P22L21: Remove one of the instances of "statistical".

P22L25: Consistency of symbol for TOA reflectance.

P23L4-5: Insert commas after "unknown" and "$\rho^{\mathrm{TOA}}$ is fixed".

P24L2: Reformat equation to fit.

---

## Referee Comment (RC3) · Anonymous Referee #2 · 29 Nov 2017

This manuscript presents a statistical method of aerosol (plus surface) retrieval from the multispectral satellite measurements, it is very well written with lucid explanations and convincing statistics. I have no objection if this paper is being published in present form. The only suggestion I have is to make it clear that "the retrieval is carried out simultaneously in all the [dark land] pixels of a granule" (page 22 line 4). The average retrieval time of "one minute per granule" only applies to the subset retrieval of dark land pixels. That being said, I do have a general comment to make about such statistical approach (another recent example is Hashimoto and Nakajima, doi:10.1002/2016JD025698). Usually, this kind of approach involves a first-guess of the retrievals, constrains like spatial smoothness, error statistics of the involved variables/processes, and an iterative numerical solver. Once better retrieval is accomplished by such method, the

question puzzles me is what is the key factor for this success – is it due to the accurate ancillary data used (like the first-guess of surface reflectance and aerosol parameters) or the power of the statistical approach (including constrains) itself? Is it possible that the non-statistical method would be as good as the statistical one if the same good a priori data (like the surface albedo) is used? Of course, applying a good first-guess is an integral part of the statistical algorithm, and I am not against using any better data if available (even the ones used for validation), but I am just not convinced that the statistical method is much better than the simple and independent non-statistical algorithm (like the MODIS Dark Target algorithm), given that much less ancillary data are required by the latter while few physics insights are added by the former. To me, getting a good estimate of the error statistics and a priori data is not easier than the retrieval itself, and the estimate of uncertainties is quite uncertain.

---

## Author Comment (AC1) · 23 Jan 2018

We thank Dr. Sayer for positive and valuable comments. Below Mr. Sayer's comments and questions are shown in boldface font followed by our replies in normal font.

**This is a very interesting and important study which provides what appears to be a better-performing way of retrieving AOD from MODIS measurements over land than the land Dark Target (DT) and Deep Blue (DB) algorithms. I have talked**

**with the authors a bit about their approach at recent meetings, and am glad to see a paper on the subject appear now. After a careful reading I had a few comments/questions which I was hoping the authors could expand upon.**

**The authors present their work as a Bayesian DT (BDT) approach, which essentially implies recasting the DT algorithm within a more formal error propagation system. As part of this statistical formalism, they also simultaneously retrieve all valid L2 pixels in a granule, which allows the use of spatial variability constraints, rather than using the independent pixel approximation, and transform much of the data into log space to avoid unphysical negative values. This is all good stuff. I think that the manuscript is written and presented well, the approach has technical merit, and the authors appreciate some nuances about DT that others often do not (e.g. the FMF is not "fraction of AOD from the fine mode" as it is in some other data sets, but "weight of the fine-mode dominated aerosol optical model").**

**Digging down, there are two other major changes: (1) the 550 nm band is also used in the retrieval (DT does not use this band) and (2) surface reflectance becomes a retrieved quantity (using the MODIS BRDF/albedo product as a prior constraint) rather than the spectral shape being an assumed quantity. These both have bigger implications, and are what I have questions about.**

**On (1), since the authors are adding this band, they must be generating new LUTs (since there is no pre-existing DT 550 nm land LUT). I may have missed it but did not see which radiative transfer code is used to generate the LUT? Is this the same as is used in the MODIS DT algorithm? And why was the 550 nm band additionally added; what happens if it is not used, is performance comparable? I know that MODIS DT and some other algorithms choose not to use this band**

**for retrievals over land, as assumed spectral/directional surface reflectance relationships don't always work so well for 550 nm as some other wavelengths.**

We actually did use the pre-existing DT 550 nm land LUT. This LUT is not directly used in the operational DT retrieval but it is distributed with the stand-alone land code (https://darktarget.gsfc.nasa.gov/reference/code). The reason for adding this band to our approach was simple: we used all pre-existing LUTs that were easily available. Our algorithm is not restricted to the four selected bands and one topic for future research is to add more bands into the algorithm and see how the results are improved.

We carried out a test in which the 550 nm band was not taken into account in our retrievals. The retrieval accuracy of our algorithm was clearly worse than in the retrievals carried out with the 550 nm band. This indicates that the 550 nm band plays an important role in the retrievals and also suggests that adding additional bands into the retrieval may still clearly improve the retrieval results. As adding new bands is a good topic for future research, we will more carefully analyse the effect of including different bands into the retrieval in the future.

**Point (2) is the bigger thing. For me, the defining characteristic of the DT algorithm is the assumption that the swIR region can be used to model reflectance in the blue and red bands, according to the relationships developed first by Kaufman and then expanded by Levy. All algorithms must make some simplifying constraining assumption about surface reflectance and this is the core of what DT is and what differentiates it from other approaches. For (almost) any sensor, when the AOD is low, the dominant over-land source of AOD retrieval error comes from surface model error (since most of the signal is surface reflection), so a retrieval's surface reflectance model is the first-order determinant of how the retrieval will behave and when it will and won't work well.**

**The BDT approach, on the other hand, retrieves surface reflectance simultaneously with AOD and FMF, using an aggregation of the MODIS BRDF product (which is itself a time-aggregated product based on atmospheric correction of MODIS imagery) and variability constraints to provide an a priori. In this sense these a priori constraints are the new surface model at the core of the algorithm and I expect the key to why it appears to work better than standard DT and DB. This is a bit more similar to e.g. the Deep Blue approach over deserts (to oversimplify, a climatology of surface reflectance obtained from the clearest 15% of scenes) or the MAIAC approach (where a time series of a number of days is built up and then surface and atmosphere are retrieved together) than it is to DT. The BDT algorithm has, unless I have misunderstood, entirely abandoned the swIR-to-visible surface model at the core of DT. All that appears to be in common are the aerosol optical models and cloud screening. This is not a criticism of the method, which appears sound. But it leads me to my main question: the BDT approach is clearly an approach which works well, but is it really correct to call it "Bayesian Dark Target", when the core feature of DT is the aspect which was discarded?**

**In my mind, it is not and it would be better to pick a different name as BDT could be misleading. The name DT conjures up the MODIS DT algorithm, and BDT likewise implies that. This is, for the reasons discussed above, something different.**

Yes, you are correct. In our algorithm, the swIR-to-visible surface model has been abandoned. We also agree with you with the name issue and therefore decided to change the name of the algorithm. We decided to call our algorithm as Bayesian Aerosol Retrieval (BAR) algorithm and have updated the manuscript accordingly.

**On an unrelated note, Equation 3 defines the posterior covariance matrix for the retrieved state. This can be used to provide pixel-level uncertainty estimates for retrieved AOD (and other quantities), a topic of much current interest. It would be interesting to compare these to the actual AOD retrieval errors against AERONET, in a statistical sense, to assess whether these are reasonable. For example, for the subset of matchups with an actual retrieval absolute error of X, is the distribution of estimated uncertainties consistent with an expectation of an error of X? (See section 3.3 of Popp et al. 2016, doi:10.3390/rs8050421 for some other example analyses looking at validating pixel-level uncertainties.) If yes, great. If not, when and where there is a mismatch between typical estimated uncertainties and typical actual errors can tell you something about which terms in your error budget are not quite right.**

We agree that it is important to evaluate the quality of the uncertainty estimates and made a figure similar to that in doi:10.3390/rs8050421, see Figure 1 for this new figure. This figures shows that the uncertainty estimates are in general overestimated for most of the AOD retrievals (small AOD values). There is room for improvement in the estimates but we think we are at least with small AOD values on the safe side as the uncertainty estimates do not give overoptimistic uncertainty estimates. The Table 8 in the manuscript basically shows the same information and from the table we can see that for large AOD values the uncertainty estimates are in general underestimated. We leave the further analysis and improvement of the error estimates as topics for future research.

**I also had a comment on the results shown in Figure 6. The high bias in Ångström exponent (AE) in both DT and BDT when the AERONET AE is low (i.e. likely cases dominated by dust) may well be because the 'coarse-dominated' aerosol model used in the retrievals assumes spherical particles, which do not model the scattering/absorption of nonspherical dust particles well. This means**

**that the phase function is simulated poorly at some angles, and the spectral dependence of absorption and extinction is incorrect. Positive AE biases are one characteristic signature of this problem. Some theoretical simulations of this are shown in Mischenko et al 1997, doi:10.1029/96JD02110; more recently, we gave (over ocean) a practical demonstration of the differences between spherical and spheroid assumptions in Lee et al 2017, doi:10.1002/2017JD027258.**

Actually, the coarse aerosol model in Dark Target over land assumes spheroid-shaped particles (for details see for example doi:10.1029/2006JD007815). Regardless of this, it is still possible that the AE biases are due to aerosol models. We thank you for the nice references that allows us to study this more. As in this manuscript we mainly describe the algorithm, we will leave the more careful analysis on effect of aerosol models for AE retrievals for a topic for future research.
* * *
[Figure]

**Fig. 1.** Histograms of the the Bayesian Aerosol Retrieval (BAR) algorithm 1-standard deviation uncertainty estimates (red lines) compared to the difference between the AERONET and MODIS BAR AOD retrievals (blu

---

## Author Comment (AC2) · 23 Jan 2018

We thank Dr. Povey for encouraging and constructive comments. Below we have included the referee's comments in boldface font and our replies below each of the comments.

[Figure]

**P4L21** **The retrieval of** $log\ x$ **rather than** $x$ **for positive variables is well documented. You retrieve** $log(1+x)$**, which I have not encountered before. Could you further discuss this choice, maybe providing references to demonstrate it's use elsewhere? I am concerned that it permits** $-1 \leq x$**. Did you specifically wish to retain the small but negative** $\tau$ **from the original Dark Target algorithm or are the 'constraints that exclude nonphysical solutions' (P5L23) hard limits that prevent this behaviour? If the latter, why not use the more common** $log\ x$ **formulation? Have you considered how hard limits distort Gaussian uncertainty estimates near those limits?**

In our optimization solution, we constrain the values of $\log(x+1)$ to non-negative values that guarantees us with non-negative AOD retrievals. As we retrieve $\log(\tau + 1)$, the prior model is written for the same quantity as well. If we retrieved $\log(\tau)$ instead, it would mean that for really small AOD values we would have really small $\log(\tau)$ values (approaching to $-\infty$ when AOD approaches 0). This in turn would require the use of really high variances (or standard deviations) for the $\log(\tau)$ to allow for higher AOD values to be retrieved. To avoid this problem of small numbers and high variances in the prior models, we decided to retrieve $\log(1 + \tau)$ instead of $\log(\tau)$. To make this clear, we added a sentence: "In practice, these constraints are implemented in the optimization algorithm." to the manuscript.

We also agree that the hard limits may distort the Gaussian uncertainty estimates but have not studied it further.

**§2**     **It's unclear from the text precisely how many pixels are processed at
once. Is it an entire granule? Processing 50,000 pixels at once would
be an impressive computational task!**

**I also recall that you only process pixels for which a DT retrieval was
produced (implicitly adopting their cloud flag), but I don't find that men-
tioned in this text**

We retrieve the same dark land, cloud-free pixels as the Dark Target algorithm, not all
pixels in a granule. We have clarified this throughout the manuscript. The maximum
amount of pixels for a MODIS aerosol granule at 10 km resolution is 27540 (204 by 135)
and in most cases only a fraction of these pixels are cloud-free, dark surface pixels.
Therefore, in practice the average amount of pixels retrieved simultaneously is about
4000 pixels. In some cases, if the amount of pixels to be retrieved has been larger than
ten thousand pixels we have divided the granule into two sub-granules and carried out
the retrievals in these sub-granules separately to make the computations faster. We
have also made it more clear that our algorithm uses the same preprocessing as the
DT by changing the sentence "BDT is a retrieval algorithm based on DT." in Section 2
to "BAR is a retrieval algorithm that uses the same aerosol models and preprocessing
of the data, such as cloud-screening, as the DT. Because the same preprocessing is
used, the BAR algorithm retrieves the same pixels as the operational DT algorithm."

**P6L17**     **Though the 50 km correlation length is widely used, you should cite
something. doi:10.1175/1520-0469(2003)060<0119:MVOTA>2.0.CO;2 is
quite common.**

Thank you for the reference. We have added a citation to this paper.

**§3.1.3** **This method contains a few surprising features. Why use blue sky albedo? Why seasonal averages? Why average the 3 closest values rather than do a bilinear or triangular interpolation? Was the technique overly sensitive to these choices (i.e. were these chosen at random and worked or did it take several attempts to find a stable solution)?**

The blue sky albedo (with weight of 0.5) was selected based on initial tests in which it resulted in the best retrieval results. In the tests, we carried out retrievals with white-sky, black-sky, and blue-sky (weight 0.5) albedos for a small amount of granules and compared the results. The differeces between the surface albedos were small but the blue-sky was the best performing one so we selected it to be used in the algorithm. We added the following sentences to the surface reflectance prior model section of the manuscript: "This selection to use the blue sky albedo was done based on a test in which we carried out retrievals with white-sky, black-sky, and blue-sky albedo based prior models. The differences between the different surface albedo types were small but the blue-sky albedo resulted in the best results when compared with the collocated AERONET AOD values."

The nearest-neighbor interpolation was selected as it is a computationally cheap and easy to implement option. As there is a surface reflection prior model with a relatively good spatial resolution, we believe that the use of some other interpolation method would not result in significantly different results.

**P8L7** **What motivated the addition of coarse mode aerosol to the continental mode? Are the Dark Target team considering removing this step from their own processing?**

The coarse mode was added to the continental mode only for convenience as our practical implementation of the algorithm requires two different aerosol models and by

doing this choice we can use the same code to process all pixels in all granules. We note that it would be a straightforward task to include the continental mode into the algorithm similarly as in the Dark Target but we think it would not significantly affect the results.

**P9L2-4** **I don't understand what you mean by 'marginalize the posterior model'. Marginalize means 'to treat as insignificant' and you use posterior model to describe the cost function, (2). I would expect one to 'minimize the posterior model', but I fail to see why that is relevant to the approximation error approach.**

In the statistical approach, by marginalization we mean that the unknown uncertainty and noise related parameters are integrated out of the posterior probability density. In our case, the posterior probability distribution is a joint conditional probability distribution of AOD, FMF, surface reflectance, and the observation noise and approximation errors given the reflectances observed by the MODIS instrument. Formally the marginalization is:

$$\pi(\tau, \eta, \rho_{\mathsf{surf}} | \rho_{\mathsf{TOA}}) = \int_e \pi(\tau, \eta, \rho_{\mathsf{surf}}, e | \rho_{\mathsf{TOA}}) \mathrm{d}e \tag{1}$$

where $\tau$, $\eta$, $\rho_{\mathsf{surf}}$, $\rho_{\mathsf{TOA}}$, and $e$ denote the AOD, FMF, surface reflectances, top-of-atmosphere reflectances observed by the MODIS instrument, and the observation noise and approximation errors, respectively. In the approximation error approach, the integration is carried out approximatively. We have added the following sentence to the manuscript to clarify this: "This means that we integrate the approximation error related variables out of the full posterior probability distribution. This is a typical approach in statistics to treat unknown nuisance parameters." For more information on marginalization of posterior probability distribution and techinques how to carry it out in practice see, for example, doi:10.1615/Int.J.UncertaintyQuantification.v1.i1.10 or
doi:10.1007/b138659

**P9L31** **'Physical' may be a better word than 'true' here as there arguably is a 'true' FMF as defined by the Dark Target algorithm, but the point is that that value doesn't always mean something in reality.**

We agree, 'true' was changed to 'physical'.

**Fig. 1** **Could the urban sites (discussed in §5.4) be displayed in a different colour?**

The figure was edited, the urban sites are now shown in a different color.

**Figs. 2&3** **For Angstrom exponent, could you use a colour bar that has grey at the centre so we can distinguish missing data from a value of 1?**

The colormap for the Ångström exponent was changed to distinguish missing data from a value of 1.

**App. B** **I broadly like this idea, and do something similar myself (though not yet in a published paper), but I'm curious about assuming the MODIS BRDF is accurate. It's a good retrieval but not without substantial uncertainty (of many forms - representational, approximation, etc.). Considering the dominance of the surface in aerosol error budgets, how accurate do you think these estimates of the approximation error are?**

We have not run a more detailed analysis on this. We also agree that the MODIS surface reflectance products may have substantial uncertainties. We assume the MODIS

surface reflectance as accurate in the approximation error computations but at the same time we use a large amount of data for constructing the approximation error model. As a large amount of data and averaging is used in the computations, it is only necessary to have unbiased (or small bias) surface reflectances in the computations to get a good approximation error model. Therefore, not all single values of the surface reflectances have to be close to correct one. The results show that the retrievals are significantly improved when we introduce the approximation error model. This shows that the models (including surface reflectance) used in the computations are of good quality to be used for this kind of modeling.

- **Several references list a URL twice. Perhaps replace the BibTeX field url with doi?**

URL problem fixed. The BibTeX field url was replaced with doi.

**I also include some proofreading recommendations.**

We thank for the recommendations. Most of the recommendations were included in the revised manuscript.

---

## Author Comment (AC3) · 23 Jan 2018

We thank the referee for the encouraging and constructive comments. Below we have included the referee's comments in boldface font and our replies below each of the comments.

**General points**

**I am somewhat concerned by the way the approximation error (§3.3 and Ap-**

[Figure]

**pendix B) is computed. In §3.2 the authors state that the mean of the measure-
ment noise PDF ($\mathbb{E}_n$) is assumed to be zero. However, it appears that this con-
straint is not applied when computing the approximation error mean ($\mathbb{E}_u$) from
comparing MODIS TOA reflectance to values simulated from AERONET AODs.
Is this correct? If it is, then this approach is making an implicit bias correction
to the MODIS L1B reflectances, based on AERONET aerosol measurements and
the retrieval's own forward model (plus the MCD43C surface reflectance) - this is
fine as it stands, although it's clearly a bit of fudge. However, the authors then
use the same AERONET measurements as validation data. It is thus not a huge
surprise that they see a significant improvement in the bias against AERONET
compared to the DT and Deep Blue products. Indeed, as the correction is com-
puted separately for different regions (the same ones used in the validation?)
and seasons, we might expect it to improve the correlation and RMS of global
and yearly comparisons of AOD or FMF vs AERONET as well. If I am correct
in my reading of how the approximation error is calculated and applied, then I
would like to see the authors provide a comparison against AERONET where $\mathbb{E}_u$
is assumed to be zero. Otherwise, a clarification of how the approximation error
is calculated is needed**

In this study, the measurement noise is always assumed to be zero-mean. By mea-
surement noise we mean the noise for example due to the instrument electronics. This
has been clarified in the revised manuscript by changing: "...the random observation
noise in MODIS observations is modelled by..." to "...the random observation noise in
MODIS observations, for example due to measurement electronics in the instrument,
is modelled by". In the computation of the approximation error statistics, we assume
the measurement noise $n$ to be small enough and set it to zero (but only in the con-
struction of the statistics). This, however, does not mean that we would neglect the
measurement noise in our retrieval but we model it and take it into account as random
noise.

We have used an independent, older AERONET dataset for constructing the approximation error statistics (evaluation of retrievals was year 2015 and the approximation error statistics data from year 2014). We have added a sentence: "...AERONET data from 2014 (one year before the test year 2015). Also, as the approximation error statistics is generated using an independent AERONET dataset, the evaluation of the algorithm will not be using the same data and therefore not result in overoptimistic results that could be possible if same datasets would be used both for modeling and evaluation of the algorithm." to Section 4 to clarify the fact that the approximation error model was based on an independent set of measurements and therefore the comparisons we carry out are fair.

Some statistics for aerosol retrievals without the approximation error model are shown in Tables 2 and 3 of the manuscript.

**The result resented in §5.2, that the retrieval performs best vs AERONET if the prior constraint of the spatial correlation of AOD and FMF are switched off, doesn't seem make sense without considering the above point, as you are then retrieving 6 parameters (four surface reflectances + AOD + FMF) from 4 measurements. Thus the results in tables 3 and 4 need further explanation.**

In our text presented in §5.2, we have changed "Globally, the best correlation between the MODIS and AERONET retrievals is observed when the approximation error model is used and spatial correlation models are turned off. These results, however, should be interpreted very carefully as they only show the global statistics. In single retrieval cases, the spatial correlation models may even play more critical role than the approximation error model. As the aerosol properties usually have clear spatial correlation we would recommend using the spatial correlation models in the retrievals." to "Globally, the best correlation between the MODIS and AERONET retrievals is observed when the approximation error model is used and spatial correlation models are turned off.

This result was unexpected as the spatial correlation models were expected on average to improve the retrieval accuracy. The results show, however, that the use of spatial correlation models does not increase the accuracy of the retrievals on average. These results, however, should be interpreted very carefully as they only show the global average statistics. In single retrieval cases, the spatial correlation models may be helpful especially in some specific scenarios or, for example, if higher spatial resolution were used. Also, the spatial correlation model parameters may play a significant role in the accuracy of the retrievals. Due to differences in local meteorology and aerosol sources, regional models for the spatial correlation may be needed to reach the best possible accuracy of the algorithm. In this study, the correlation model parameters were not based on a thorough analysis of aerosol properties correlation structures, and only a global correlation model was used. As the aerosol properties usually have clear spatial correlation we would recommend using the spatial correlation models in the retrievals. "

**I think that the fact that the best correlation, bias, fraction within $EE_{DT}$ and RMS error all correspond to the configuration where spatial correlation is disabled, but approximation error is enabled, is simply due to the regional/seasonal bias correction against AERONET implicitly applied by the approximation error methodology. Again, there is nothing inherently wrong with doing such a correction, but you cannot then pretend that AERONET is an independent source of validation data.**

As explained above, the AERONET data used to construct the approximation error model was from an independent dataset (different year) than the AERONET data used for algorithm evaluation.

**Furthermore, tables 3 and 4 show that applying the correlation constraints don't actually improve the results vs AERONET, even if the approximation error is dis-**

[Figure]

**abled. This would seem to imply that the correlation constraints aren't improving the retrievals at all. I would be surprised if it turned out that these constraints don't actually improve the retrieval in many cases, but I feel that a concrete example is needed, rather than the vague assurances given at the end of §5.2.**

We cannot say that the correlation constraints are not improving the retrievals at all. There may be cases in which the correlation helps and there may be cases in which the correlation makes the retrievals worse. On the other hand, it seems that the use of the spatial correlation model does not make the retrieval accuracy significantly worse. Also as it is clear that aerosol properties have spatial correlation structures we decided to include this possibility in our algorithm. It could be, for example, that in higher-resolution retrievals the correlation constraints would be more helpful. Also, we have not carefully analyzed the optimal correlation model parameters but just selected some reasonable values for the correlation models and the retrievals could be improved if more realistic correlation models were used. As this manuscript is mainly an algorithm description, we leave the further analysis of the spatial correlation models to future research. As stated above, we have modified the text in the manuscript to make it more clear that here the spatial correlation model did not seem to be the key factor in improved AOD retrievals.

**Also, am I right in thinking that if both the approximation and spatial correlation errors are switched off, the retrieval effectively assumes the prior surface reflectance values from the MCD43C product are correct (i.e. the retrieval doesn't move from the a priori values)?**

The optimization algorithm is based on gradient information and therefore it depends on the sensitivity of our posterior distribution which variables are optimized in the retrieval. If both the approximation error model and spatial correlations are switched off, it does not mean that the retrieval algorihm would assume the prior surface reflectance values

as correct but optimizes them similarly as AOD and FMF. Of course, in this case there are more unknowns than observations and therefore the solution in general may not be unique. In this case, the retrieved values also may depend on the initial point selected for iteration. In our case, we always select the most likely values according to our prior models as initial points for iteration.

**I agree with the point made in an earlier comment by Dr Sayer regarding the name of the algorithm. Clearly, given that the processing done by the authors shares the cloud-clearing, look-up tables (and thus aerosol models) etc as the DT algorithm, it would seem fair enough to call the resulting product as the Bayesian Dark Target product, but the retrieval algorithm itself could be described as opposite in approach to the NASA DT algorithm, as BDT does away with both the independent pixel and spectrally correlated surface reflectance assumptions which form the basis of the DT (and Deep Blue) approaches (while introducing its own assumptions about spatial correlations in aerosol properties). My feeling is that the authors are in danger of "under selling" the algorithm, as, without going through the details of the algorithm, it could appear to just be DT with pixel-by-pixel uncertainty estimates.**

Based on Dr Sayer's and referees' comments, we have changed the name of the algorithm to Bayesian Aerosol Retrieval (BAR) algorithm and revised the manuscript accordingly.

**In a similar fashion, I'd like to see the authors move away from the (in my opinion misleading) use of the term fine-mode fraction to describe the fraction of the AOD due to the fine mode. Perhaps "fine mode AOD fraction" is a better term? At the very least, the definition of what is actually meant by FMF needs to be stated up-front.**

The fine mode AOD fraction may also be misleading because in Dark Target and our algorithm the fine mode fraction is actually the fraction of top-of-atmosphere reflectance due to fine aerosol part of the model. We define the FMF as the 'fine aerosol model weighting' in our manuscript. Furthermore, we added the following sentence to Section 2: "It should be noted that in DT algorithm, the FMF is actually the weighting coefficient for the TOA reflectances due to fine aerosol model, and do not necessary represent the true concentration fraction of the fine-mode aerosol."

**Finally, though I appreciate that adding in an additional product into their validation would be a bit much, the authors don't seem to be aware of the MAIAC MODIS product, which (as Dr Sayer noted) in some ways has more in common with their approach than DT.**

We have now mentioned the MAIAC algorithm with references to doi:10.1029/2010JD014985 and doi:10.1029/2010JD014986 in the introduction.

**Specific points**

**P1L1: Suggest you reword to "...(BDT) algorithm for the retrieval of aerosol optical depth over land from MOderate Resolution..."**

Reworded as suggested.

**P1L18: Reword to "...particles may be hazardous to human health when inhaled..."**

Corrected as suggested.

[Figure]

**P1L21: Add a comma after "predictions".**

Corrected as suggested.

**P2L4: I'd describe DT as the primary operational algorithm used to retrieval aerosol, not just "an algorithm".**

Was changed to: "The primary operational algorithm to retrieve aerosol properties, such as the aerosol optical depth (AOD), is the Dark Target (DT)..."

**P2L7: Reword to "...DT algorithm is the brightening effect, whereby an increased amount of aerosol over dark..."**

Changed as suggested.

**P2L19: Remove "for example".**

Removed.

**P2L24-L27: The sentences describing the Deep Blue algorithm don't scan well at all.**

The sentences were changed to: "DB is used for over land aerosol retrievals and it was developed especially for retrievals over bright-reflecting surface. The capability of retrieving aerosol properties over bright-reflecting surfaces is useful, for example, in retrieving dust properties over deserts. Regardless of the bright-reflecting surface capabilities, DB does not carry out retrievals over snow or ice."

**P2L34: Please provide a reference/justification for the statement that taking advantage of spatial correlations in aerosol properties can improve retrieval results. I note that your own results (Section 5.2, Tables 3 and 4) don't actual support this statement (although I'd be surprised if it wasn't true!)**

We agree that the statement may have been too strong. We have changed "Often, however, aerosol properties have strong spatial correlations. Taking advantage of the spatial correlations of aerosol properties in the retrieval can, in many cases, improve the accuracy of the retrieved parameters." to "Often, however, aerosol properties have strong spatial correlations (Anderson et al, 2003). Modeling and taking advantage of the spatial correlation structures of aerosol properties in the retrieval may therefore, in some cases, improve the accuracy of the retrieved parameters."

**P4L8: Why did you choose the listed bands in particular? Was just because these were the ones for which look-up tables were available?**

The reason for selecting the listed bands in particular was that the lookup-tables were already easily available for these bands. The algorithm is not restricted to these bands only and the use of all possible data is expected to improve the retrieval accuracy. The downside of using more bands, however, would be increased computational costs. We consider retrievals with more bands as a topic for future research.

**P4L21: As with the other referee, Dr Povey, I am also confused by the $\tau = log(\tilde{\tau} + 1)$. I understand retrieving AOD in log space (not only do you avoid the problem of negative AODs, but the observed distribution of AODs is much closer to log-normal than normal), but where does the "+1" come from? Also, as Dr Povey notes, $log(\tau + 1)$ is defined for $\tau > -1$, and thus doesn't prevent negative optical-depth values as you state in the text.**

In our optimization solution, we constrain the values of $\log(x+1)$ to non-negative values that guarantees us with non-negative AOD retrievals. As we retrieve $\log(\tau + 1)$, the prior model is written for the same quantity as well. If we retrieved $\log(\tau)$ instead, it would mean that for really small AOD values we would have really small $\log(\tau)$ values (approaching to $-\infty$ when AOD approaches 0). This in turn would require the use of really high variances (or standard deviations) for the $\log(\tau)$ to allow for higher AOD values to be retrieved. To avoid this problem of small numbers and high variances in the prior models, we decided to retrieve $\log(1 + \tau)$ instead of $\log(\tau)$. To make this clear, we added a sentence: "In practice, these constraints are implemented in the optimization algorithm." to the manuscript.

**P4L22: The retrieval using log reflectance is also new to me. Can you elaborate on why this was chosen? In particular, doesn't this complicate the use of the estimates of the measurement noise of the MODIS reflectances? And, again the "+1" needs explaining.**

The use of log-scale reflectances were selected based on a test where we compared the AOD retrievals with and without log-scale. The use of log-scale resulted in slightly better retrieval accuracy so we decided to use it. In this setting, the measurement noise (+1) is normal-distributed in log-scale (lognormal distributio). This means that it is slightly more probable that the true reflectance is higher than the observed reflectance than it would be lower than the observed reflectance. As the reflectances in AOD retrievals are generally quite small this seems to be more realistic assumption and results in slightly improved retrievals. The "+1" was used for the same reason as for the AOD, please see the reply regarding the use of $\log(\tau + 1)$.

**P4L25: Please explicitly define the symbols used for expected value vectors and covariance matrices here.**

[Figure]

Definitions added.

**P5L6: Eq (2) needs reformatting so that it fits within the margins.**

Equation reformatted.

**P6 (§3.1.1 and 3.1.2): I don't agree with your approach in presenting the Prior models and their covariance matrices in particular. For a start, the values for $\sigma^2_{\text{nugget}}$ make up a small fraction of the total variance (3rd sig. fig. for AOD and 2nd sig. fig. for FMF) and thus are largely irrelevant. Furthermore, I think it be clearer if you were define a correlation matrix and variance matrix separately, which are then combined to give you your covariance. The off-diagonal elements of a covariance matrix are combination of the variance of the variables concerned (where an increased value corresponds to a less tight constraint) and correlation between them (where a increased value implies a tighter constraint).**

We highly appreciate your opinion about the presentation of the covariance matrices. This form of presentation, however, is similar to the ones we have seen in the literature (for example the reference we have used doi:10.1029/2009JD013765). To make it easier for the readers to compare our model with others from the literature we would like to leave the presentation of the covariance matrices as it is now. **P6L16-18: The**

**authors need to acknowledge that an inherent weakness of their scheme, and the use of a spatial correlation constraint in particular, is that it will always result in the smoothing of sharp features in the aerosol field (such as smoke plumes). There is no way you will be able to "correctly" retrieve the AOD of a near-source aerosol plume when assuming a correlation in aerosol AOD and FMF over 50 km.**

The sentence was changed from "This selection was made to let the neighbouring pixels have relatively high spatial correlation but on the other hand to allow for certain features such as smoke plumes to be retrieved correctly and not be smoothed out." to "This selection was made to let the neighbouring pixels have relatively high spatial correlation but on the other hand to allow for certain features such as smoke plumes to be retrieved as well as possible and not be smoothed out too much."

The use of this type of spatial prior information does not necessarily mean that the retrieved AOD distribution is always very smooth but it means that there is a high probability for the AOD distribution to be smooth. In this type of prior model, AOD distributions with rapid changes, for example due to smoke plume, still have a non-zero probability. In cases where the observations really strongly suggest that there are rapid changes in AOD it is possible that these rapid changes are seen in the retrieved distributions as well.

**P7L14: Please provide some reasoning behind the choice of the blue sky albedo, and the 0.5 weighting coefficient in particular.**

The blue sky albedo (with weight of 0.5) was selected based on initial tests in which it resulted in the best retrieval results. In the tests, we carried out retrievals with white-sky, black-sky, and blue-sky (weight 0.5) albedos for a small amount of granules and compared the results. The differeces between the surface albedos were small but the blue-sky was the best performing one so we selected it to be used in the algorithm. We added the following sentences to the surface reflectance prior model section of the manuscript: "This selection to use the blue sky albedo was done based on a test in which we carried out retrievals with white-sky, black-sky, and blue-sky albedo based prior models. The differences between the different surface albedo types were small but the blue-sky albedo resulted in the best results when compared with the collocated

AERONET AOD values."

**P7L29: I know Eq (5) is fairly ubiquitous in aerosol and cloud remote sensing, but a reference for its derivation would be nice.**

References to Chandrasekhar Radiative Transfer (1960) and doi:10.1109/TGRS.1986.289617 were added.

**P8L11: How are the look-up tables corrected for surface elevation? Is this an additional parameter in the table?**

The same approach as in the Dark Target algorithm is used for correcting for the surface elevation. In this approach, the lookup-tables are directly corrected for and therefore no additional variables for the tables are not needed. The method is based on adjustment of the Rayleigh optical depth at different surface elevations. For more information on the surface elevation method see for example: MODIS Dark Target ATBD or Fraser, R. H., Ferrare, R. A., Kaufman, Y. J., Mattoo, S. (1989). Algorithm for Atmospheric Corrections of Aircraft and Satellite Imagery. NASA Technical Memorandum 100751. Greenbelt, MD USA, NASA Goddard Space Flight Center.

**P9L8-9: Change to "We model the approximation error $u$ as spatially, but not spectrally, uncorrelated, meaning the correlations between MODIS bands are taken into account".**

Corrected as suggested.

**P9L10: Delete the "the" before "spatial and season variations", Also, place a**
**comma after "Similarly".**

Corrected as suggested.

**P9L19: Remove "the" before "±30".**

Removed.

**P9L25: Change to "In order to evaluate the near real time..."**

Changed.

**P10L14: Remove "the" before "DB retrievals".**

"the" removed.

**P11L13: The use of "oversmoothed" here irks me somewhat. For a start, how does one define what is "oversmoothed"? I feel the authors should be up-front and acknowledge that there is some evidence in Fig. 3 that their algorithm is smoothing the AOD field to a certain degree - in particular, I am looking at the thin smoke plume slightly above the smoother, wider area of smoke in the middle of the image. There is a clear linear feature in both the true-colour and DT images, which largely absent from the BDT results.**

Thank you for the remark. We have changed the sentence: "Regardless of the spatial correlation model used for AOD in BAR, the plume is not oversmoothed and shows similar details as the DT retrieval." to "The use of spatial correlation model for AOD in

BAR can be seen as slight smoothing of the plume details when compared to the DT retrieval."

**P11L14-17: The authors present some circumstantial arguments to suggest that the BDT FMF in Fig. 3 is superior to that provided by DT, and I agree that some of the features present in the DT results are almost certainly artefacts. However, due to the spatial constraints applied to the BDT algorithm, it is always going to produce a more pleasing looking, smoothly varying field than an independent pixel approach like DT. This doesn't mean that we can conclude that, for the scene in question, we can say which product is quantitatively more accurately. Again, some acknowledgement of the fact that some real features might be smoothed-out in the BDT product should be included.**

The following sentence was added to the manuscript: "It should be noted, however, that the spatial correlation model for FMF may in some cases result in too smooth FMF fields that are unrealistic, for example in cases of smoke plumes, reducing the accuracy of the retrievals in these cases."

**P12L2: Insert a comma after "algorithms".**

Comma inserted.

**P12L4: Insert a comma after "Figure 4".**

Comma inserted.

**P12L8: Change to "...was carried out, based on the DT algorithm QA flag, which**

**is designed to discard ...**"

Changed.

**P13L6: Change to "Visual inspection shows the BDT retrievals..."**

Changed.

**P21L28: I have never heard of anyone trying to use spatial correlation constraints on the surface land reflectance. This doesn't seem to make sense as the land surface is generally pretty anisotropic at all spatial scales. Do the authors mean spectral correlation instead?**

We do mean spatial correlation. It is possible to write a suitable spatial correlation model for the land surface. It would not necessary be a correlation model implying smoothly varying surface but, for example, the anisotropic nature of the surface could be encoded into the model. As stated in the manuscript, this is a topic for future research.

**P22L1: I don't believe the claim that the signal-to-noise ratio of MODIS reflectances is too low to allow accurate aerosol retrievals. The reasons that aerosol products have usually been done on a courser spatial scale than native instrument resolution are all to do with dealing with residual cloud contamination and surface anisotropy (and reducing product size and computational cost).**

We agree with you in signal-to-noise ratio issue. We have changed the future research item text to "High-resolution retrievals. In high-resolution pixel-by-pixel retrievals, the anisotropic and non-smooth surface reflectance, and residual cloud contamination are

major sources of uncertainties and may lead to poor retrieval accuracy. BDT takes into account the spatial correlations of aerosol properties and this may make the algorithm more tolerant to higher uncertainties. Therefore, the use of BDT would especially improve the high-resolution (3 km) aerosol retrievals."

**P22L4: Due to the inclusion of the AERONET based approximation error term, the scheme already has a kind of data-fusion with AERONET!**

We rather think it is more an aerosol model calibration than data-fusion as the approximation error model is constructed using independent, older AERONET dataset. But you are correct, if the AERONET data used in the approximation error model were collocated in time with the MODIS retrievals it could be thought of a simple data-fusion with AERONET.

**P21/P22: I have an additional thought for future development of the algorithm: Is there any reason the approach couldn't be applied over the ocean?**

There are no restrictions to use the algorithm over ocean as long as there is some information about the surface. Especially over ocean there could be clear benefits of using spatial correlation model for the surface reflectance. In Dark Target over Ocean algorithm, the ocean surface reflectance depends on the wind speed. A similar surface model could also be used with our retrieval algorithm as well (Dark Target surface reflectance as prior mean, for example). We thank the referee for this idea and have listed the over ocean retrieval as a topic of future work in the conclusions.

**P22L21: Remove one of the instances of "statistical".**

Removed.

**P22L25: Consistency of symbol for TOA reflectance.**

By $\rho^{TOA}$ we denote the simulated TOA reflectances and by $\rho^{TOA,MODIS}$ the true MODIS observed TOA reflectances. We have clarified this by changing "...given the observed TOA reflectances..." to "...given the true MODIS instrument observed TOA reflectances..."

**P23L4-5: Insert commas after "unknown" and "$\rho^{TOA}$ is fixed".**

Commas added.

**P24L2: Reformat equation to fit.**

Equation reformated.
* * *

---

## Author Comment (AC4) · 23 Jan 2018

We thank the referee for the good and positive comments and questions. Below we have included the referee's comments in boldface font and our replies below each of the comments.

Referee comment:

**This manuscript presents a statistical method of aerosol (plus surface) retrieval from the multispectral satellite measurements, it is very well written with lucid explanations and convincing statistics. I have no objection if this paper is being**

[Figure]

**published in present form. The only suggestion I have is to make it clear that "the retrieval is carried out simultaneously in all the [dark land] pixels of a granule" (page 22 line 4). The average retrieval time of "one minute per granule" only applies to the subset retrieval of dark land pixels. That being said, I do have a general comment to make about such statistical approach (another recent example is Hashimoto and Nakajima, doi:10.1002/2016JD025698). Usually, this kind of approach involves a first-guess of the retrievals, constrains like spatial smoothness, error statistics of the involved variables/processes, and an iterative numerical solver. Once better retrieval is accomplished by such method, the question puzzles me is what is the key factor for this success – is it due to the accurate ancillary data used (like the first-guess of surface reflectance and aerosol parameters) or the power of the statistical approach (including constrains) itself? Is it possible that the non-statistical method would be as good as the statistical one if the same good a priori data (like the surface albedo) is used? Of course, applying a good first-guess is an integral part of the statistical algorithm, and I am not against using any better data if available (even the ones used for validation), but I am just not convinced that the statistical method is much better than the simple and independent non-statistical algorithm (like the MODIS Dark Target algorithm), given that much less ancillary data are required by the latter while few physics insights are added by the former. To me, getting a good estimate of the error statistics and a priori data is not easier than the retrieval itself, and the estimate of uncertainties is quite uncertain.**

We thank the referee for the constructive comments and good questions.

Throughout the manuscript, we have added a mention of dark land pixels (instead of all pixels of a granule) to clarify that only the pixels with dark land are retrieved.

The use of statistical approach for retrieval as such does not make the algorithm per-

fom better from the conventional approaches. Actually, in many cases the construction of the retrieval algorithm using a statistical approach leads to very similar practical solution of the retrieval problem than the deterministic approach. The biggest difference in statistical approach, in many cases, is the way of thinking (modeling all unknowns as random variables). This statistical way of thinking allows more straightforward processing of, for example, uncertainties related to the models used in the algorithm. Also this type of statistical approach directly allows the use of all possible data (prior information, measurements, more realistic noise and uncertainty models) that may improve the retrievals. We think that in this case the use of more realistic models and uncertainties, better surface information, and the use of observation data (more bands) in the retrieval are the keys for better performance of the algorithm. For more general introduction to statistical inversion methods, please see for example doi:10.1007/b138659